# RF Transceiver for the Multi-Mode Radar Applications

**DOI:** 10.3390/s21051563

**Published:** 2021-02-24

**Authors:** Jae Kwon Ha, Chang Kyun Noh, Jin Seop Lee, Ho Jin Kang, Yu Min Kim, Tae Hyun Kim, Ha Neul Jung, Sang Hwan Lee, Choon Sik Cho, Young Jin Kim

**Affiliations:** School of Electronic and Information Engineering, Korea Aerospace University, Goyang 10540, Korea; yo0600@kau.kr (J.K.H.); Ckn@kau.kr (C.K.N.); dl1717@kau.kr (J.S.L.); hojin321@kau.kr (H.J.K.); kimym4269@kau.kr (Y.M.K.); flowerwell@kau.kr (T.H.K.); hnjung@kau.kr (H.N.J.); imsh0404@kau.kr (S.H.L.); youngjinkim@kau.ac.kr (Y.J.K.)

**Keywords:** multi-mode radar, FMCW, CW, pulse radars, RF Transceiver

## Abstract

In this work, a multi-mode radar transceiver supporting pulse, FMCW and CW modes was designed as an integrated circuit. The radars mainly detect the targets move by using the Doppler frequency which is significantly affected by flicker noise of the receiver from several Hz to several kHz. Due to this flicker noise, the long-range detection performance of the radars is greatly reduced, and the accuracy of range to the target and velocity is also deteriorated. Therefore, we propose a transmitter that suppresses LO leakage in consideration of long-range detection, target distance, velocity, and noise figure. We also propose a receiver structure that suppresses DC offset due to image signal and LO leakage. The design was conducted with TSMC 65 nm CMOS process, and the designed and fabricated circuit consumes a current of 265 mA at 1.2 V supply voltage. The proposed transmitter confirms the LO leakage suppression of 37 dB at 24 GHz. The proposed receiver improves the noise figure by about 20 dB at 100 Hz by applying a double conversion architecture and an image rejection, and it illustrates a DC rejection of 30 dB. Afterwards, the operation of the pulse, FMCW, and CW modes of the designed radar in integrated circuit was confirmed through experiment using a test PCB.

## 1. Introduction

Radar is an electronic device that detects and discriminates targets regardless of weather or day light condition. In various environments that cannot be easily identified by human eyes, radars are being used to effectively detect targets, especially in the case of military applications, evolving into complex, multi-period surveillance and reconnaissance radars that perform anti-drones, and civilian and scientific missions. It has been applied to the automotive and aerospace fields and is continuously being advanced.

The FMCW radar is mainly used as a long-range radar (LRR) by adjusting the frequency modulation bandwidth and frequency modulation time [1,2]. On the other hand, the UWB radar is used as a short-range radar (SRR) because it shows a high range resolution [3,4]. However, the FMCW radar is difficult to use as an LRR if the frequency modulation bandwidth and frequency modulation time are not sufficiently secured, and there is a disadvantage in that high output power must be provided. In addition, as the UWB radar should secure a wide bandwidth, it may affect other communication services and shows a disadvantage of limiting the maximum output power. As such, each radar has been usually implemented separately due to very different characteristics such as output power, bandwidth, and center frequency.

As a result, the number of radar sensors increases depending upon various applications, making the structure of the radar system very complex, bulky, and costly. Therefore, a technology for implementing a multi-mode radar with one IC is required, and research has been conducted to realize a radar transmitter supporting FMCW/UWB mode with a single chip [5]. However, radar transceivers that support three modes such as pulse, FMCW, CW modes have not been studied so far. Therefore, in this work, we propose multi-mode radar transceivers for pulse mode for long-range target detection and for FMCW and CW modes for detection and identification of short-range targets integrated in a single chip [6].

The conventional radar transceivers show some constraints. For the transmitters, the TX blocker in which the transmitted signal directly enters the receiver called self-interference (SI) and the LO leakage in which the LO signal is immediately radiated occur. This creates DC offset in the receiver, contaminates the received signal and makes it impossible to accurately identify the targets. Therefore, it is necessary to suppress LO leakage by removing the DC offset generated in the baseband of the transmitter.

For the receivers, a radar system uses a Doppler frequency of several Hz to several kHz to detect targets. A very low frequency flicker noise is created between SiO${}_{2}$ and the channel of MOSFETs comprising the receiver circuitry, leading to worsening the S/N ratio of the receiver [7]. Furthermore, as DC offset occurs due to SI and LO leakage of the transmitted signal, the noise figure is deteriorated, and the long-range detection is greatly degraded due to contamination of the received signal. Therefore, there is a need for the receiver that can improve noise figure and effectively suppress DC offset [8].

Section 2 shows the design requirements of transceivers for multi-mode radar systems. Section 3 explains the proposed transceiver, and describes the principle of suppressing LO leakage by applying an 8-bit current DAC to the baseband of the transmitter [9]. We also describe the proposed receiver, by analyzing the direct conversion and a double conversion structures with image rejection and explain about DC offset rejection [10,11,12,13,14]. In Section 4, simulation results of the designed transceiver is shown. After that, we describe the design and fabrication of a test PCB for measuring transceivers and radar tests for multi-mode applications. Finally, in Section 5 we conclude the measurement and radar test results of the transceiver for the proposed multi-mode radar system.

## 2. Design Requirements for Multi-Mode Radar Transceivers

In this work, three different modes—pulse, FMCW, and CW—are considered for the multi-mode radar transceivers. Design specifications are decided based on these three modes.

### 2.1. Pulse Radar

In pulse radar, the maximum detection distance ($R_{max}$) depends on the transmit power and pulse width (τ) which is the reciprocal of bandwidth (*BW*). The maximum detection distance according to the pulse width (τ) is obtained as follows. The Friis formula as shown in (Equation 1) can be used for calculating $R_{max}$ where the minimum received power ($P_{R,min}$) is obtained from Equation (Equation 2). Equation (Equation 3) derives $R_{max}$ using Equations (Equation 1) and (Equation 2) where $P_{T}$ is the transmit power; $P_{R}$ is the received power; $G_{T}$ and $G_{R}$ are the transmit/receive antenna gains, respectively; σ is RCS of the target; $NF_{tot}$ is the total noise figure of the receiver; $kT_{e}$ is the thermal noise power density; and BW is the bandwidth which is the reciprocal of the pulse width [15]. Figure 1 shows the calculated maximum detection distance ($R_{max}$) according to transmit power, pulse width (τ) and noise figure of 12.9 dB which is derived for the design specification later in Section 2.2.
(1)$$P_{R}=\frac{G_{T}G_{R}\sigma \lambda^{2}}{{\left(4\pi \right)}^{3}R^{4}}P_{T}$$
(2)$$P_{R,min}=NF_{tot}\cdot kT_{e}\cdot BW\cdot SNR_{min}$$
(3)$$R_{max}=\sqrt[{4}]{\frac{G_{T}G_{R}\sigma \lambda^{2}}{{\left(4\pi \right)}^{3}kT_{e}\cdot BW\cdot SNR_{min}}\cdot \frac{P_{T}}{NF_{tot}}}$$

As the pulse width increases, radar can detect farther targets as shown in Figure 1. Furthermore, if the transmitter can emit high transmit power, it can detect farther targets. The maximum distance to target of 322 m can be obtained using τ of 2 ms and transmit power of 13 dBm using Equation (Equation 3) where λ can be computed from RF carrier frequency of 24 GHz used for this work. Therefore, the output power of the transmitter is designed to be greater than 13 dBm in this work.

### 2.2. FMCW Radar

The purpose of the multi-mode radar in consideration is to detect targets from a long distance in the pulse radar mode, and to continuously detect and identify targets in a short range in FMCW and CW modes when a moving target approaches from the very long distance. Therefore, to design the transceiver of the multi-mode radar system in FMCW mode, the maximum detection distance needs to be confirmed by analyzing the transmit power of the transmitter, the noise figure of the receiver, the frequency modulation bandwidth, and frequency modulation time.

Radar systems can detect targets through Doppler shift appearing in several kHz. However, in transceivers implemented in CMOS, flicker noise causes a significant problem in the Doppler shift. If the flicker noise is severe, the radar’s long-distance detection is rapidly deteriorated because the Doppler frequency of several kHz cannot be detected. Therefore, the maximum noise figure allowed to detect the Doppler frequency can be determined using sensitivity analysis as shown in Equations (Equation 4) and (Equation 5) which are rewritten from Equations (Equation 1) and (Equation 2), respectively.

The detectable minimum input signal level $P_{in,min}$ can be expressed as shown in Equation (Equation 4). $P_{in,min}$ of −115.1 dBm is calculated by substituting the transmit power ($P_{T}$) of 13 dBm, the transmit/receive antenna gains of 25 dBi, the free space loss (LOSS) of −94 dB, and the drone’s Radar Cross Section (RCS) of 0.001 into Equation (Equation 4). If SNR is set to 16 dB, the maximum allowable noise figure of 12.9 dB can be obtained by substituting 1 kHz Doppler shift (BW) into Equation (Equation 5).
(4)$$P_{in,min}=P_{T}+G_{T}+G_{R}+LOSS+RCS$$
(5)$$P_{in,min}=-174dBm/Hz+10log\left(BW\right)+SNR_{out}+NF$$

Figure 2 shows the maximum detectable distance ($R_{max}$) calculated according to the transmit power and the noise figure using Equations (Equation 1)–(Equation 3) for FMCW mode [15]. In Figure 2, the x-axis represents the noise figure, the y-axis represents the transmit power, and the z-axis represents the distance. It can be seen that as the transmit power increases and the noise figure decreases, the detectable distance increases. More calculated results for maximum detectable distance are tabulated as shown in Figure 2. When the maximum detection distance in FMCW mode is calculated using the noise figure of 12.9 dB and the transmission power of 13 dBm obtained earlier, a detection distance of 84.4 m can be obtained.

In order to secure the range resolution in FMCW mode, the frequency modulation bandwidth (*B*), frequency modulation time (*T*), and beat frequency need to be calculated and set as the design requirements of the radar. Beat frequency ($\Delta f_{R}$) and the distance to the target (*R*) in the FMCW radar can be expressed as shown in Equation (Equation 6) and range resolution can be computed using a specific beat frequency of $\Delta f_{R}=250$ kHz which is hardly affected by flicker noise [14,16]. Figure 3 shows the target range calculation according to frequency modulation time (*T*) and frequency modulation bandwidth (*B*).
(6)$$R=c\cdot \frac{T\Delta f_{R}}{2B}$$

The beat frequency is also calculated according to the distance and velocity of targets by setting B to 75 MHz and T to 50 s specified from Table 1. As flicker noise appears in the range of 0 to several kHz, it is confirmed that targets with a distance of 10 to 100 m and a velocity of 10 to 100 km/h can be detected by measuring beat frequency of the 30 kHz to 300 kHz, which is less affected by the flicker noise. The detailed calculation is summarized in Table 1. In this work, only UP chirp has been used due to the limitation of test equipment.

### 2.3. CW Radar

The CW radar can detect the Doppler frequency of an approaching or retreating target, determining the presence or absence of a target and the target’s velocity. Because CW radar can only know the velocity of the target, it must be used together with the FMCW radar to detect the distance to target. In addition, as the FMCW radar has multiple-target ambiguity, a CW waveform is usually added after one period of the FMCW waveform to distinguish the components for pure Doppler shifts and find a frequency that matches the intrinsic Doppler component to resolve the multiple-target ambiguity. In this work, CW radar is applied only for the verification of transceiver operation.

### 2.4. Design Specification Summary

The design requirements for the proposed multi-mode radar transceiver is summarized. The minimum input signal, maximum noise figure, maximum detection distance, frequency modulation bandwidth in FMCW mode, and detection distance according to modulation time are confirmed in Table 2. It can be seen that the RCS is set for a drone [17], and the transmit/receive antenna gains are also obtained by horn antenna manufactured for this work. For a reasonable NF requirement (7∼13 dB for overall receiver), the antenna gain will need to be quite high based on a wide BW leading to sufficient radar detection performance. Moreover, the thermal noise power density at normal temperature is −174 dBm/Hz. As it has a bandwidth of 1 kHz, the maximum thermal noise power density is −144 dBm/Hz. Afterwards, design requirements are verified by measuring the transceiver and testing the radar.

## 3. Design of the Proposed Transceiver

Figure 4 is the block diagram for the proposed multi-mode radar transceiver. In the transmitter side, LO leakage in RF TX output is suppressed by devising a current DAC. In the receiver side, a chopping frequency is generated from LO block used for mixers for avoiding the flicker noise, leading to lower noise figure. In addition, LO leakage and TX blocker rejection are suppressed using the 2nd mixer. The transceiver is composed of transmitter (TX), receiver (RX), buffer to input external LO signal and auxiliary circuits such as VCM generator to generate common mode voltage, current generators (IREF), bandgap-reference (BGR), low drop regulator (LDO), and an SPI. The TX is composed of analog baseband (ABB), up conversion mixer, and power amplifier (PA). Moreover, RX consists of ABB, mixers, and low noise amplifier (LNA). BGR generates internal bias current and voltage, and LDO supplies 1.2 V of voltage to each block. Furthermore, SPI controls all functions of the blocks. The supply voltage is 3.3 V, and an external LO signal is applied to the mixer of the transmitter and the first mixer of the receiver. The baseband input frequency of the transmitter is 5 MHz, and it radiates 24.005 GHz signal at the output of the transmitter. The receiver receives the frequency of 24.005 GHz + $F_{d}$ and obtains $F_{d}$ from the baseband output of the receiver. At this time, in order to reduce the flicker noise of the receiver, the LO frequency of the first mixer of the receiver was designed in a structure in which 40 MHz (Chopper frequency) was added to 24 GHz. In this section, the structure of the proposed transmitter and the principle of LO leakage suppression are described. The structure of the proposed receiver is then described compared with direct conversion receiver. In particular, as the proposed receiver can reduce the influence of flicker noise, a lower noise figure can be obtained compared to the direct conversion receiver.

### 3.1. Architecture of Direct Conversion Transmitter

The transmitter is designed in a direct-conversion architecture. This architecture guarantees high linearity compared to other types of transmitters. However, if the LO leakage is generated by the DC offset originated in the baseband, and the signal power of the LO leakage can become larger than the transmit power, it is impossible for receiver to obtain the desired signal. This problem saturates the receiver as well as the transmitter, directly affecting the far-field detection performance of the radar system. Therefore, there is a need for a method that can effectively remove the DC offset. An 8-bit current DAC is devised for removing the DC offset as shown in Figure 5. In addition, as a double balanced mixer can structurally suppress the LO leakage signal, a conventional passive mixer is employed for the proposed transmitter [8]. The PGA uses a complementary two stage op-amp to secure sufficient gain and phase margin for the transmitter [18].

#### 3.1.1. LO Leakage Suppression through DAC

Figure 6 shows the principle that LO leakage occurs in a direct conversion transmitter. A signal including DC offset enters into the I/Q phase and is mixed with the LO signal, going to the $RF_{OUT}$. $RF_{OUT}$ can be expressed as shown in Equation (Equation 7).
(7)$$\begin{array}{cc} RF_{OUT} & =\left(V_{DO\_I}+A_{BB}cos\omega_{BB}t\right)A_{LO}sin\omega_{LO}t+\left(V_{DO\_Q}+A_{BB}sin\omega_{BB}t\right)A_{LO}cos\omega_{LO}t\\ \; & =A_{BB}A_{LO}sin\left(\omega_{BB}+\omega_{LO}\right)t+A_{LO}\left(V_{DO\_I}sin\omega_{LO}t+V_{DO\_Q}cos\omega_{LO}t\right) \end{array}$$
where $A_{LO}\left(V_{DO\_I}sin\omega_{LO}t+V_{DO\_Q}cos\omega_{LO}t\right)$ occurs due to DC offset and causes LO leakage. This does not occur when there is no DC offset.

Figure 7 shows the circuit diagram for CDAC where the current can be sequentially adjusted by multiples of 300 nA from LSB to MSB. Current can be adjusted up to 19.2 A, and the input values of the P and N nodes are reversed through the MSB. For example, if 1 is set to MSB, the PMOS of the P node is turned off and the PMOS of the N node is turned on.

If DAC is connected to the P and N nodes of the PGA, respectively, and the current according to the 8-bit CODE is supplied to the P node and the current comes out of the N node, a DC voltage is obtained by the product of the equivalent resistance $R_{eq}$ of the PGA and the current $I_{CODE}$ according to the CODE of the DAC as in Equation (Equation 8). The voltage $V_{Calibration}$ as much as the offset can be generated, and DC offset is removed. Through SPI, we adjust the CODE of 8 bits, check the LO leakage on the spectrum analyzer, and adjust the CODE in the direction that LO leakage decreases.
(8)$$V_{Calibration}=R_{eq}(R_{1}∥|R_{2})\times I_{CODE}=V_{DC\_Offset}=V_{P}-V_{N}$$

#### 3.1.2. Power Amplifier (PA) Design

The maximum transmit power ($P_{T}$) is set to 13 dBm in design requirement. For this purpose, a power amplifier and a drive amplifier with matching networks are designed as in Figure 8.

As the PA must supply a high power to the antenna, the power gain is compensated additively through the DA. Therefore, inter-stage matching between DA and PA is also required, and the DA output and the PA input are matched using a transformer [19,20].

Because PA consumes a lot of current, the transistor size becomes very large, and as the transistor size increases, the parasitic capacitance also increases. In particular, as a leakage current is generated by the parasitic capacitor $\left(C_{gd}\right)$ between the drain and the gate, and the gain and efficiency of the power amplifier decrease. Therefore, it is designed to cancel the leakage current by applying a cross-coupled capacitor (CCC) as shown in Figure 9. If $i_{CCC}$ and $-i_{Cgd}$ are the same, the leakage current is canceled effectively.

The impedance of the maximum power transfer was confirmed through the power amplifier design. As the impedance seen from the drain of MOSFET used for PA at a specific bias is 62.68 − j50.22, conjugate matching was employed for maximum power transfer. At this time, the small signal impedance of the PA was also checked and matched so that the $S_{22}$ viewed from the output of the PA could come out below −10 dB, and the simulation result of the maximum output power of about 12 dBm was confirmed.

At this time, the gate bias voltages of DA and PA having the same circuit configuration but different bias are designed to 480 mV and 650 mV, respectively. In Table 3, design parameters for transmitter are summarized using the design philosophy introduced in Section 3.1.

### 3.2. Architecture of Receiver

#### 3.2.1. Conventional Receiver Architecture

Figure 10 shows the conventional receiver architecture using the direct conversion structure where 3-stage inductively degenerated differential LNA [7] and down mixer are usually adopted. As the direct conversion receiver directly down-converts the RF signal to the baseband, it is highly affected by the flicker noise in the baseband. Therefore, the double conversion is alternatively employed for this work.

Figure 11a shows that SNR decreases due to the image signal generated in direct down-conversion, and Figure 11b shows the LO leakage and transmission signal generated by the parasitic components of the mixer. This illustrates that DC offset is generated due to TX blocker directly inserted from the transmitter. Therefore, we propose a double conversion receiver and add a DC Offset rejection circuit to suppress the flicker noise.

#### 3.2.2. Proposed Receiver Architecture

The proposed receiver consists of a 3-stage LNA, 1st down mixer, 2nd down mixer, and baseband circuitry such as the high-pass filter. The degenerated inductive differential configuration is employed for LNA as drawn inside Figure 10. In order to reduce the flicker noise, improve noise figure, and remove DC offset, three techniques are applied to the 2nd down mixer.

#### 3.2.3. Proposed Double Conversion Receiver

Figure 12 shows the proposed double conversion architecture and SNR in each block. The received signal is down-converted to the baseband through the IF band less affected by flicker noise. Compared to the direct conversion architecture of Figure 10, the receiver gain increases as much as the gain of the 2nd down mixer with Transimpedance Amplifier (TIA) included, and thus a lower noise figure can be obtained.

#### 3.2.4. LO Leakage and TX Blocker Rejection Technique

The LO leakage generated by mixer and LNA parasitics, and the TX blocker where the transmitted signal directly enters the receiver port, create a DC offset in the baseband of the receiver. This DC offset contaminates the received signal from the target and should be removed as much as possible. Figure 13 shows the proposed DC offset cancellation principle. Here, the signal path from node 1 to node 2 becomes the main path, and the feedback path from node 2 to node 4 operates as an LPF. The whole transfer function from IF_I to BB_I becomes an HPF as shown in the right top of Figure 13. The cut-off frequency of HPF can be set to 3 kHz or 160 kHz by adjusting the capacitance. When the LO leakage indicated in blue enters the mixer’s input node (node 1), the LO leakage is down-converted through the mixer and appears as a DC at the 2nd mixer’s output node. The signal to be detected is marked in red at the baseband node (node 2) where the cut-off frequency of the low-pass filter (LPF) of the feedback loop is set lower than the detected signal so that only the DC component passes the LPF. Thereafter, the fed-back DC signal is up-converted to the LO frequency and transmitted to node 4. If the phase of the DC signal delivered to node 4 through the switch becomes out of $180^{\circ }$, LO leakage can be effectively eliminated at the output node (node 2).

Figure 14 is a diagram for the mixer and TIA used in the 1st mixer and the 2nd mixer where a passive mixer configuration is employed to increase linearity. In the quadrature structure, when the mixer is driven with a 50% duty cycle, there is an overlapped period in which the I and Q-paths are turned on at the same time as shown in Figure 15. In this case, current flows simultaneously through the I- and Q-paths. The current flowing between the I- and Q-paths passes through the channel of the switching transistor, creating additional flicker noise. For this reason, the mixer was devised to be driven with a 25% duty cycle to prevent the I- and Q-paths from being turned on at the same time, and the generation of such additional flicker noise was suppressed [10].

When the mixer adopts a 25% duty cycle for LO injection, a waveform for the positive I phase ($g_{LOIP}$) having a duty of [−T/8, T/8] in Figure 15 can be expressed by Equation (Equation 9) where $\eta_{0}$ means the duty ratio. Therefore, one can find $g_{LOIN}$, $g_{LOQP}$, and $g_{LOQN}$.
(9)$$\begin{array}{c} g_{LOIP}=\eta_{0}+\sum_{n=1}\left(\frac{2}{n\pi }\sin\left(n\pi \eta_{0}\right)\cos\left(n\omega t\right)\right) \end{array}$$

$g_{LOQN}\cdot g_{LOIP}-g_{LOQP}\cdot g_{LOIN}$ can be expressed as shown in Equation (Equation 10).
(10)$$\begin{array}{cc} \; & g_{LOQN}\cdot g_{LOIP}-g_{LOQP}\cdot g_{LOIN}=2\left[\eta_{90}\frac{2}{\pi }\sin\left(\pi \eta_{0}\right)\cos\left(\omega t\right)+\frac{2}{\pi }\sin\left(\pi \eta_{90}\right)\sin\left(\omega t\right)\eta_{0}\right.\\ \; & +\frac{2}{\pi }\sin\left(\pi \eta_{90}\right)\sin\left(\omega t\right)\frac{1}{\pi }\sin\left(2\pi \eta_{0}\right)\cos\left(2\omega t\right)\left.-\frac{1}{\pi }\sin\left(2\pi \eta_{90}\right)\cos\left(2\omega t\right)\frac{2}{\pi }\sin\left(\pi \eta_{0}\right)\cos\left(\omega t\right)\right]\\ \; & \;\;\;\;\;\;\;\;\;\;\;\;\;\;\;\;\;\;\;\;\;\;\;\;\;\;\;\;\;\;\;\;\;\;\;\;\;\;\;\;\;\;\;\;\;\;\;\;\;\;\;\;\;\;\;\;\;\;\;\;\;\;\;\;\;\;\;\;\;\;\;\;\;\;\;\;\;\;\;\;\;=\frac{2\sqrt{2}}{\pi }\cos\left(\omega t-\frac{\pi }{4}\right) \end{array}$$

Using Equation (Equation 10), the conversion gain of the switching cell of the mixer is calculated, and the final gain considering the TIA can be expressed by Equation (Equation 11), and the gain is determined by the on−resistance of the mixer and the feedback resistance of the TIA.
(11)$$\begin{array}{cc} \frac{V_{X,1}\left(t\right)-V_{X,2}\left(t\right)}{V_{RFP}\left(t\right)-V_{RFN}\left(t\right)}\; & =\frac{2\left[V_{RFP}\left(t\right)g_{LOQN}\cdot g_{LOIP}+V_{RFN}\left(t\right)g_{LOQP}\cdot g_{LOIN}\right]}{V_{RFP}\left(t\right)-V_{RFN}\left(t\right)}\\ \; & =\frac{\sqrt{2}}{\pi }\cdot \frac{\cos\left(\omega_{IF}t+\frac{\pi }{4}\right)}{\cos(\omega_{RF}t)} \end{array}$$

Furthermore, the total gain can be expressed by Equation (Equation 12).
(12)$$\begin{array}{cc} Total\;Gain & =\frac{V_{X,1}\left(t\right)-V_{X,2}\left(t\right)}{V_{RFP}\left(t\right)-V_{RFN}\left(t\right)}\times \frac{V_{OUT}\left(t\right)}{V_{X}\left(t\right)}=-\frac{\sqrt{2}}{\pi }\times \frac{R_{feed}}{R_{on}}\left[1-\frac{1}{A_{o}}\left(1+\frac{R_{feed}}{R_{on}}\right)\right] \end{array}$$

In the meantime, for mixer’s noise analysis, as shown in Figure 16, it can be expressed as an equivalent model. Internal resistance $R_{S}$ is expressed as a resistance of $R_{S}/2$ at each input node in consideration of differential input, and noise generated by OP AMP is expressed as $V_{n,op}$ as an input referred noise source. The noise source at each resistance is expressed in voltage, the switching operation occurs in the mixer, and the RF signal is converted into an IF signal.

NF is found using 4 noise sources (①∼④) represented as shown in Figure 17. As it can be expressed as $NF=\frac{\bar{{V^{2}}_{out1}}}{{A^{2}}_{V,total}}\cdot \frac{1}{4kTR_{S}}$, $\bar{{V^{2}}_{out1}}$ and ${A^{2}}_{V,total}$ are found using Equations (Equation 12) and (Equation 13).
(13)$$\begin{array}{cc} \bar{{V_{out1}}^{2}} & =\left(\frac{\;4\sqrt{2}}{\pi }KTR_{S}+\frac{\;4\sqrt{2}}{\pi }KTR_{on}\right)\times {\left(\frac{R_{feed}}{\frac{\sqrt{2}}{\pi }\left(R_{on}+R_{s}\right)}\right)}^{2}\left[1-\frac{1}{{A_{o}}^{2}}{\left(1+\frac{R_{feed}}{\frac{\sqrt{2}}{\pi }\left(R_{on}+R_{s}\right)}\right)}^{2}\right]\\ \; & +4KTR_{feed}+\bar{{V_{n,op}}^{2}}{\left(\frac{\left(R_{on}+R_{s}\right)\frac{\sqrt{2}}{\pi }+R_{feed}}{R_{s}\frac{\sqrt{2}}{\pi }+R_{on}\frac{\sqrt{2}}{\pi }}\right)}^{2} \end{array}$$

Substituting the Equations (Equation 12) and (Equation 13) into $NF=\frac{\bar{{V^{2}}_{out1}}}{{A^{2}}_{V,total}}\cdot \frac{1}{4kTR_{S}}$, the final noise figure can be expressed by Equation (Equation 14).
(14)$$\begin{array}{cc} NF & =1+\frac{R_{on}}{R_{S}}+\frac{\sqrt{2}}{\pi }\frac{{\left(R_{on}+R_{s}\right)}^{2}}{R_{S}R_{feed}}\frac{1}{\left[1-\frac{1}{{A_{o}}^{2}}{\left(1+\frac{R_{feed}}{\frac{\sqrt{2}}{\pi }\left(R_{on}+R_{s}\right)}\right)}^{2}\right]}\\ \; & +\bar{{V_{n,op}}^{2}}\frac{\pi }{4\sqrt{2}KTR_{S}}{\left(1+\frac{\sqrt{2}}{\pi }\frac{\left(R_{on}+R_{s}\right)}{R_{feed}}\right)}^{2}\frac{1}{\left[1-\frac{1}{{A_{o}}^{2}}{\left(1+\frac{R_{feed}}{\frac{\sqrt{2}}{\pi }\left(R_{on}+R_{s}\right)}\right)}^{2}\right]}\\ \; & \approx 1+\frac{R_{on}}{R_{S}}+\frac{\sqrt{2}}{\pi }\frac{{\left(R_{on}+R_{s}\right)}^{2}}{R_{S}R_{feed}}+\bar{{V_{n,op}}^{2}}\frac{\pi }{4\sqrt{2}KTR_{S}}{\left(1+\frac{\sqrt{2}}{\pi }\frac{\left(R_{on}+R_{s}\right)}{R_{feed}}\right)}^{2} \end{array}$$

As described in Equation (Equation 14), when the mixer is driven with a 25% duty cycle, there is no period in which the I- and Q-paths are turned on at the same time, so the flicker noise does not affect the mixer noise performance. The final noise figure is only represented by the internal resistance of the input source ($R_{S}$), the on−resistance of the mixer ($R_{on}$), and the feedback resistance of the TIA ($R_{feed}$). Therefore, the noise figure for mixer with 25% duty cycle is significantly improved compared to that with a 50% duty cycle [10,11,12]. The 1st mixer operates with 50% duty cycle used for pulse mode, on the contrary, the 2nd mixer operates with 25% duty cycle used for FMCW and CW modes. PRF for the pulse is not affected by this proposed scheme.

Summarized design parameters are illustrated in Table 4 in considering the design philosophy explained in Section 3.2.

## 4. Simulation & Measurement Results

### 4.1. Simulation

#### 4.1.1. Transmitter Simulation

600 mV DC is applied to $V_{P}$ of the PGA for the transmitter, and DC inputs varying from 600 mV to 620 mV are applied to $V_{N}$ as shown in Figure 7, and a DC offset is created by amount of $V_{P}-V_{N}$ in simulation. Figure 18 shows simulated LO leakage according to $V_{P}-V_{N}$. When DC offset is created by making 10 mV difference between the $V_{P}$ and $V_{N}$ nodes, LO leakage of about 0 dBm occurs. In design stage, the PGA was designed to provide −6–24 dB gain using 6 dB gain step. The PGA simulation confirms this performance accurately. At this time, the PGA gain is set to 24 dB, and the resistances of $R_{1}$ and $R_{2}$ set to 1 kΩ and 16 kΩ, respectively. Therefore, $R_{eq}$ is about 1 kΩ, and if a current of about 10 A is passed from the DAC, the voltage as much as DC offset can be generated. The red line is the simulation result of suppressing the LO leakage using CDAC, and it can be seen that the LO leakage can be maintained lower around −23 dBm using CDAC.

In addition, the maximum output power of the transmitter is simulated. Through the simulation, a maximum output of 12 dBm was obtained using the PA circuitry. Figure 19 confirms that the maximum gain of PA becomes stable according to the CCC value, and through this simulation result, the CCC value was determined as 48 fF. In addition, the passive up mixer provides −6 dB gain in simulation. The total current consumption of the transmitter results in 86 mA where the PA consumes 60 mA, the DA does 12 mA and the baseband circuitry does 14 mA in simulation.

#### 4.1.2. Receiver Simulation

Figure 20 shows the P1dB simulation result of the receiver. The RF frequency was set to 24 GHz, the LO frequency was set to 23.96 GHz, and RF power was used as a variable. The solid black line represents the input, the red circular symbol represents the LNA output, and the blue triangle symbol represents the output of the mixer.

Figure 21 shows the result of gain simulation with the IF frequency as a variable. The solid black line represents the input, the red circular symbol represents the LNA output, and the blue triangle symbol represents the output of the mixer. The gain of the 3-stage LNA is about 30 dB, and the gain of the 1st down mixer is about 15 dB.

Figure 22 shows the result of the transfer function simulation of IF-stage. The black solid line represents the input, the red dotted line represents the output of the 2nd down mixer when DCOC (DC Offset Cancellation) mode is turned off, and the blue triangle symbol and the green inverted triangle symbol represent the output of the 2nd down mixer when DCOC mode is turned on. The gain is obtained about 19 dB, and high-pass filter (HPF) cut-off frequencies are achieved about 160 kHz and 3 kHz.

Figure 23 shows of simulation results to check the blocker P1dB and DC rejection characteristics of the proposed receiver. The black solid line is the blocker input signal, and the red solid line symbol indicates the power of the DC component from the IF output. The blue dotted symbol indicates the DC component from the IF output by the third order term. Filled symbols indicate when DCOC mode is turned on, and empty symbols indicate when DCOC mode is turned off. Blocker OP1dB was set to 0 dBm, which is judged not to contaminate the desired signal in IF output, and it was confirmed that the blocker P1dB is improved by about 13.5 dB compared to the case where DCOC mode was turned off. In addition, it can be seen that the DC rejection effect of about 30 dB is achieved at the input less than −30 dBm.

For comparison with direct conversion receiver, the blocker test is performed. Figure 24 shows the comparison result of the block test between the proposed double conversion receiver and a general direct conversion receiver. First, looking at the blocker P1dB, the blocker IP1dB of about 17.5 dB is improved in the proposed receiver compared to the direct conversion receiver. This means that even at 17.5 dB higher than the transmit power, the received signal is not contaminated. In addition, the DC offset removal that may occur due to Tx blocker and LO leakage is improved by about 35 dB. However, for inputs above −30 dBm, this effect gradually decreases.

Figure 25 shows the noise figures of the proposed double conversion receiver and the direct conversion receiver. The red circular symbol represents the noise figure and the blue diamond symbol represents the gain. An empty symbol indicates a direct conversion receiver, and a filled symbol indicates a proposed receiver. The proposed receiver has an additional gain in the 2nd down mixer. An image rejection and DCOC loop are added to lower noise by about 30 dB compared to direct conversion at 100 Hz, which is susceptible to flicker noise. Looking at the simulation result, it can be seen that the noise figure is significantly reduced below 1 MHz.

### 4.2. Measurement

The length from the connector port to transceiver pads can vary according to the test PCB type fabricated for measurement. In order to verify a circuit in the millimeter wave band, a board that can guarantee performance in a high frequency band, modeling of wire bonding, and an appropriate impedance matching circuit are required. Therefore, a ceramic-based RO4003C substrate was used for PCB fabrication, and after modeling the wire bonding and matching circuit using a 3D CAD tool, the S-parameter was confirmed through simulation and then the final fabrication was carried out.

Figure 26 shows the modeling of the double bond wire carried out for connecting the transceiver pads to the connector port using a 3D EM-simulator, and Figure 27 shows the Q-factor and inductance of the modeled double bond wire. The Q-factor of bond wire was confirmed to be about 60 in the 24 GHz band, and the inductance was confirmed to be about 800 pH. The voltage and ground bond wires were modeled in the same way, and inductances of 1 nH and 700 pH were confirmed, respectively. In the meantime, in the circuit design stage, the design was carried out by considering the inductance of the bond wire to the RF input, output and all voltage sources. As the Q-factor and inductance of the bond wire have a great influence on the RF input and output of the IC, it is important to design it close to the ideal condition. Therefore, the length of the bond wire is reduced as much as possible and the Q-factor is designed as large as possible, so that it is designed close to the ideal inductor.

Figure 28 shows a stub circuit designed including bond wire for PCB fabrication. The signal pads we double wire bonded, and all stub circuits including wire bonding were modeled and applied to the IC to confirm the S-parameters. In addition, in order to prevent frequency shift due to errors in the actual manufacturing process, a pad that can be additionally wire-bonded to the stub circuit is designed so that the frequency can be changed appropriately.

Figure 29 shows the simulation result of S-parameter of transceiver including stub circuit from port 1 and port 2 as seen in Figure 28. At 24.25 GHz, $S_{22}$ is −22 dB, $S_{11}$ is −27 dB, and $S_{21}$ is −0.9 dB, confirming that the circuit operates normally at 24 GHz. Considering the frequency shift according to parasitic capacitance and corner variation of the IC in the actual production stage, the working frequency was adjusted to 24.25 GHz in the design stage.

The transceiver for the CMOS multi-mode radar system was fabricated in a TSMC CMOS 65 nm process as shown in Figure 30, and the total area of the die takes up 3.54 mm × 3.54 mm including the test LO circuit. A 3.3 V supply voltage was used and the measurement was carried out after wire bonding the IC to the PCB.

Figure 31 illustrates the fabricated PCB integrating the transceiver chip and radar test environment. The size of the manufactured PCB is 100 mm × 100 mm, the matching circuit of each transceiver was exposed to the side of the PCB, and the total length of the matching circuit was reduced to 17 mm.

### 4.3. Transceiver Measurement Results

#### 4.3.1. Transmitter Measurement Results

For the transmitter measurement, the PGA gain of the baseband was set to 24 dB, and at the input frequency of 30 MHz, the input power of the I phase was set to −20 dBm, and the input power of the Q phase was also set to −20 dBm. LO power is also set to 10 dBm. The transmitter consumes about 86 mA of current.

Figure 32 shows the measured frequency response of the transmitter. As it was designed by applying the direct conversion and the image rejection, the image rejection is confirmed about 21 dB, and the 3 dB bandwidth is measured 70 MHz.

Figure 33a shows LO leakage according to 8-bit CDAC code. The optimal code was confirmed by adjusting the reference current of the CDAC to 20 A and 10 A, and Figure 33b is the result of the output spectrum of the transmitter by applying the optimal code. By optimizing the CDAC code, it was possible to suppress LO leakage of about 30 dB as expected.

The measurement result of the maximum output power of the transmitter is shown in Figure 34. Through the measurement, a maximum output of 8.8 dBm was measured which is bit lower than simulation. The lower output power may be originated from the imperfect matching between DA and PA, and inaccurate bonding wire modeling.

#### 4.3.2. Receiver Measurement Results

The receiver was measured by setting RF input power of −50 dBm and LO power of 5 dBm, and Figure 35 shows the result of measuring the bandwidth of the receiver. The black graph is the measurement result by setting the front-end and baseband of the receiver to the maximum gain of 66 dB, and the red graph is the measurement result by lowering the gain of the front-end by 10 dB. In both cases, a 3 dB bandwidth of 2 GHz was measured.

Figure 36 shows the P1dB measurement result of the receiver. In the high gain column of Figure 36, by setting the receiver gain to 68.5 dB, IP1dB of −60 dBm were obtained. In the low gain column of Figure 36, with the gain set to 2.6 dB, it is found that −13 dBm of IP1dB were measured.

Figure 37 is the measured result the frequency response of HPF. In a radar system, as the received power decreases as the distance to the target increases, the decrease of the received power according to the distance must be compensated for through HPF. Therefore, the slope of HPF was set to a maximum of 40 dB/dec, and the measurement results according to turn on and off of a total of three DCOCs are shown. Graph ① is the measurement result with only the DCOC of RF TIA on, and shows the cut-off characteristic of 20 dB/dec and 10 kHz. In the graph ②, two DCOCs were turned on, and the cut-off characteristics of 30 dB/dec and 20 kHz were confirmed. Finally, graph ③ was measured with three DCOCs turned on, and showed cut-off characteristics of 40 dB/dec, 20 kHz, and it was confirmed that the Doppler frequency can be detected properly.

Figure 38 shows the measurement result by the cut-off frequency of DCOC, and it was confirmed that the cutoff frequency can be adjusted from a minimum of 10 kHz to a maximum of 250 kHz considering the CW and FMCW modes. Furthermore, DC rejection of up to 40 dB is confirmed as is Figure 38.

Figure 39 is measured result of the PGA gain of the receiver where the input power is −42 dBm, the LO frequency is 23 GHz, and the LO power is 5 dBm. At this time, the measurement was performed by setting the gain step of the PGA to 6 dB each. From the measurement results, a maximum gain of 68.5 dB was obtained.

Figure 40 compares the noise figures of a direct conversion receiver and a proposed double conversion receiver. The black graph is the noise figure of the direct conversion receiver, and the red graph is that of the double conversion receiver. The proposed receiver provides an additional gain in the 2nd down mixer. By adding image rejection and DCOC loop, a lower noise figure compared to the direct conversion receiver at 100 Hz, which is vulnerable to flicker noise, was confirmed through measurement. From the measurement results, it is confirmed that the proposed double conversion at 15 MHz shows a lower noise figure.

Figure 41 shows the receiver’s LO leakage measurement. The LO leakage of the receiver is measured with the LO power radiated to the input terminal of the receiver through the LNA by applying the LO input power to the 1st down mixer. When the LO input power is 9 dBm and −1 dBm, the LO leakage is measured as −16 dBm and −28 dBm, respectively.

Figure 42 compares the measurement result of the transceiver’s S-parameter with the simulation. In the S-parameter simulation, after checking the output impedance by setting the corner variation of the transceiver to Slow, Fast, and Typical Corner, respectively, it was set as the input impedance of the matching circuit to confirm $S_{11}$. The transmitter confirms $S_{11}$ of about −11 dB at 24 GHz, and the receiver confirms $S_{11}$ of about −10 dB at 24 GHz through simulation.

### 4.4. Radar Test

The fabricated test PCB with the proposed transceiver chip was used for radar test in multi-modes. Figure 43 shows the measurement environment for the proposed transceiver in a real radar test. The IF of the transmitter is set to 5 MHz, the LO frequency is set to 24 GHz, and the Doppler frequency or the beat frequency is received along with 24.005 GHz at the receiving end, and the measurement was performed by checking with a spectrum analyzer. A hum target is approached to check its movement. The Doppler frequency was measured using the CW mode and the speed of the target was calculated to confirm the measurement accuracy. In addition, the position of the target at rest was confirmed through the beat frequency using the FMCW mode.

#### 4.4.1. CW Mode

The transmitter’s IF is set to 5 MHz and the LO frequency is set to 24 GHz, so the output power of 8.8 dBm is measured at 24.005 GHz as shown in Figure 44. Before radiating the output power from the transmitter, the code of the CDAC was adjusted to find the code that suppresses LO leakage as much as possible. The LO leakage was set to −30 dBm or less and the signal power at 24.005 GHz was set to 8.8 dBm for radar test.

Figure 45 shows the measurement results of an incoming target and an outgoing target. It can be seen that the Doppler frequency rises by about 270 Hz for the incoming target, and the speed of the target is calculated from the Doppler frequency formula, which is 1.7 m/s. In addition, it can be seen that the Doppler frequency decreases by about 180 Hz for the target moving away, and the speed of the target is 1.13 m/s from the Doppler frequency formula. This is similar to people’s average walking speed of 1.4 m/s, so it can be seen that the CW mode works appropriately.

#### 4.4.2. FMCW Mode

When measuring the FMCW mode radar, a large frequency modulation bandwidth and a short modulation time are required to ensure sufficient distance resolution. Therefore, the IF of the transmitter was modulated from 0 to 70 MHz with a time of 50 s, and the measurement was performed. The beat frequency against the distance was measured to check whether the FMCW mode operates properly.

As FMCW mode can only use upstream frequency modulation in the current test environment as shown in Figure 46, speed information cannot be obtained, and a test was conducted to check the distance at which the target is located through the bit frequency according to the distance to the target. First aid kit made of steel with high RCS were placed at about 15 m and 20 m away to check the frequency on the spectrum and converted it into distance to confirm the operation of the FMCW mode.

As a result of calculating the distance to the target by substituting the beat frequency of the stationary target into Equation (Equation 6), the calculated values of 15 m and 20 m were obtained when the beat frequency was 140 kHz and 186.67 kHz, respectively, as shown in Figure 47. Consistently, it was possible to confirm whether the FMCW mode also works properly.

#### 4.4.3. Pulse Mode

As in Section 2.1, the maximum detection distance varies according to the transmit power and pulse width. Therefore, in the pulse radar test, the square wave was confirmed by spectrum analyzer by appropriately adjusting the transmit power and the distance to the stationary target. Through this, it was confirmed whether the pulse radar operates properly.

Figure 48 shows the pulse radar test environment. The square wave was confirmed through the spectrum analyzer by changing the distance while the drone was fixed at a constant height. When the duty cycle of the pulse is 50%, it appears as the sum of odd-order harmonics in the spectrum, so the measurement was performed by setting the transmit power to −20 dBm, the pulse width to 2 ms and pulse repetition frequency (PRF) to 250 Hz.

In Figure 49, it can be seen that odd-order harmonics of 250 Hz appear, and through this, it can be seen that the pulsed radar operates properly.

Table 5 summarizes the measurement results of the proposed transceiver compared with other works. In particular, the direct conversion receiver was mainly compared, bypassing the 2nd down mixer of the proposed receiver. Finally, the radar test results are summarized in Table 6. The proposed radar transceiver is built in a test PCB to verity the multi-mode operation such as pulse, FMCW, and CW modes. If this transceiver is employed with radar signal processing system, more accurate operation for various mode radar can be evaluated precisely.

## 5. Conclusions

In this paper, we described the design and measurement of a transceiver for multi-mode radar and performed the radar test in pulse, FMCW, and CW modes. As the LO leakage occurs in the transmitter, the image rejection is applied and the DC offset generated in the baseband is removed by using an 8-bit current DAC (CDAC) of the baseband to suppress LO leakage. As a result, the result of LO leakage rejection of 37 dB was obtained at 24 GHz. In order to improve the flicker noise and the noise figure, the proposed receiver uses a double conversion structure to down-convert through an intermediate frequency where the influence of flicker noise is relatively small, and thereby reduce flicker noise. In addition, by applying the double quadrature structure, it improves a noise figure by 3 dB through image signal removal, and is designed to distinguish when the target is moving away or when it is moving closer. Finally, it is designed to have an isolation of about 30 dB in the middle frequency band by properly suppressing DC Offset that may occur due to LO leakage and TX blocker signals. As a result, it was possible to measure a noise figure of at least 12 dB at 100 kHz and 15 dB at 100 Hz, which was improved by 20 dB compared to that of about 35 dB in the direct conversion modes receiver.

Pulse, FMCW, and CW mode measurements were performed for radar tests, and the CW mode was used to detect the Doppler frequency incoming or outgoing targets. It succeeded in detecting the distance to the target, and it was possible to check whether the CW and FMCW mode radar operate properly. 

## Figures and Tables

**Figure 1 sensors-21-01563-f001:**
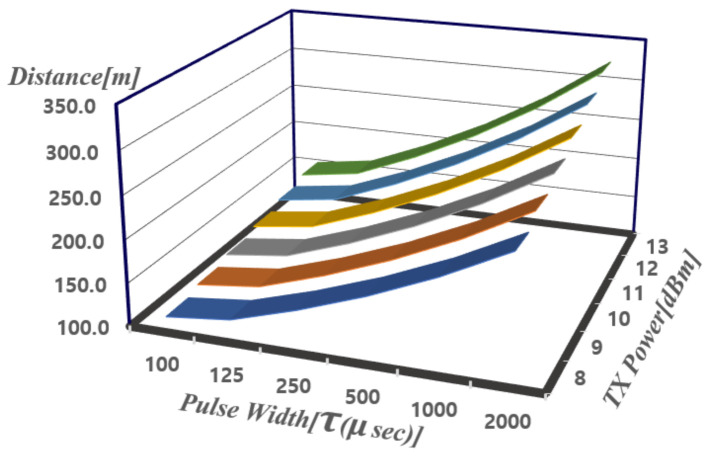
Radar detection distance ($R_{max}$) according to transmit power and pulse width (τ).

**Figure 2 sensors-21-01563-f002:**
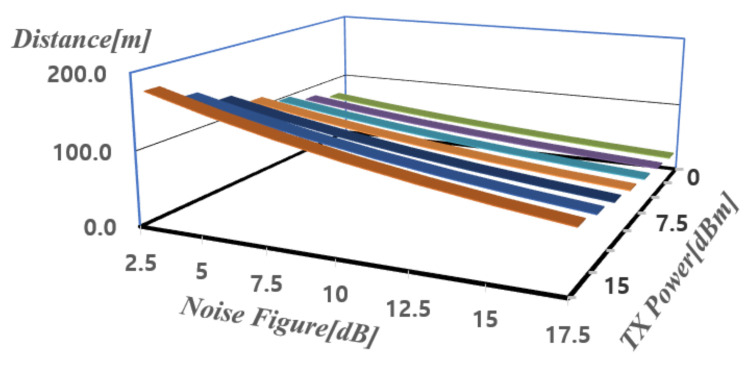
Calculation of maximum detection distance for FMCW radars.

**Figure 3 sensors-21-01563-f003:**
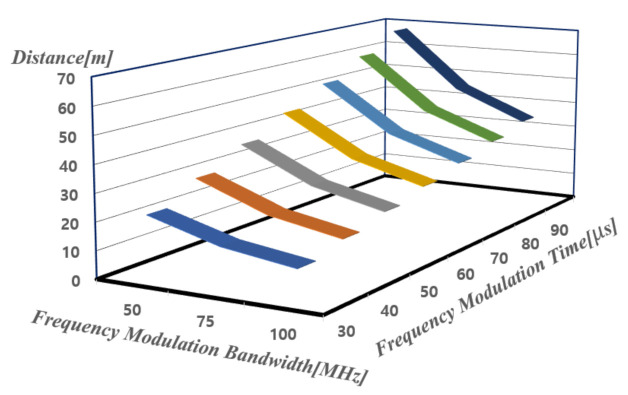
Target range calculation according to frequency modulation time (*T*) and frequency modulation bandwidth (*B*) with $\Delta f_{R}=250$ kHz.

**Figure 4 sensors-21-01563-f004:**
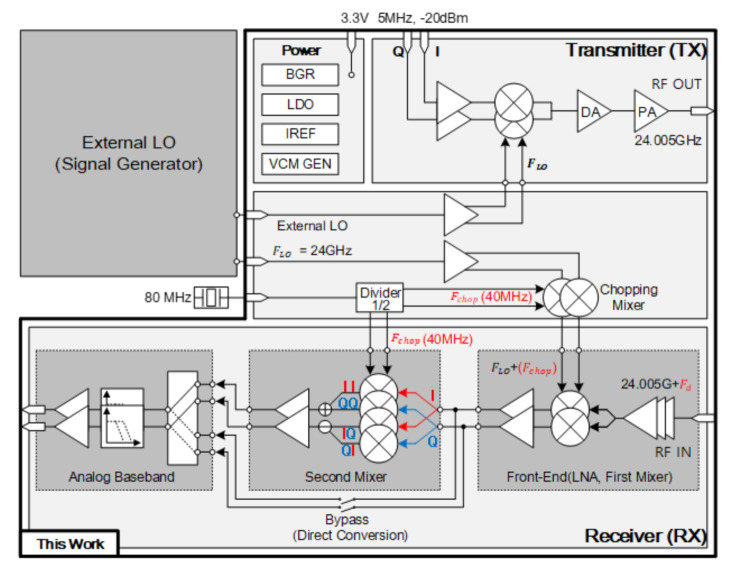
Block diagram of the proposed multi-mode radar transceiver.

**Figure 5 sensors-21-01563-f005:**
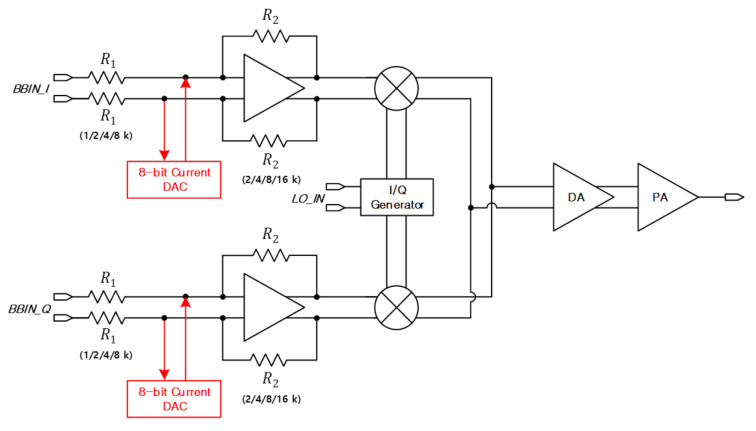
Block diagram of direct conversion structure transmitter.

**Figure 6 sensors-21-01563-f006:**
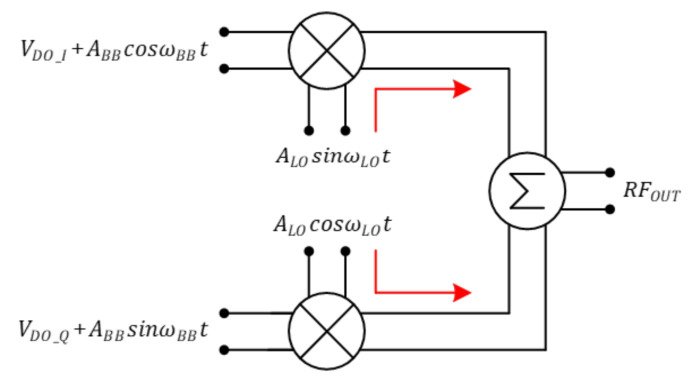
LO leakage generation in direct conversion transmitter.

**Figure 7 sensors-21-01563-f007:**
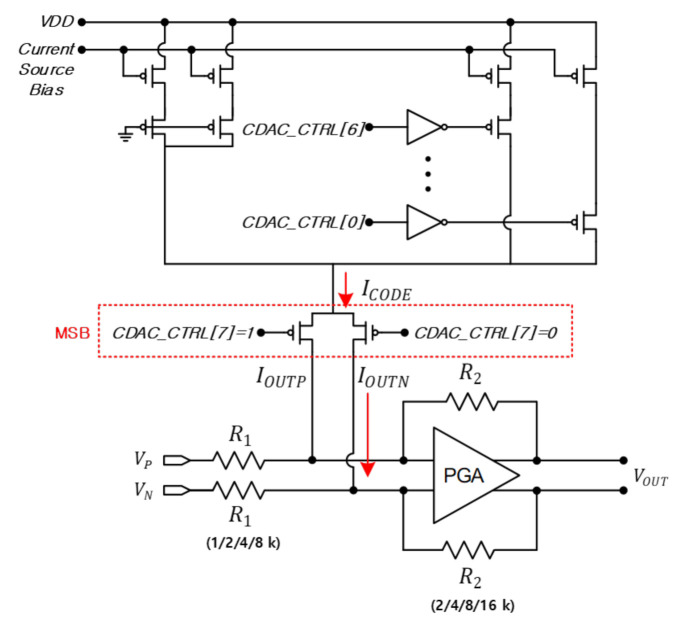
CDAC circuit diagram.

**Figure 8 sensors-21-01563-f008:**
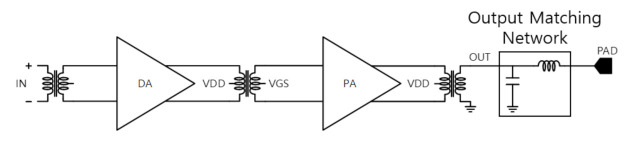
PA and Drive Amplifier (DA) block diagram.

**Figure 9 sensors-21-01563-f009:**
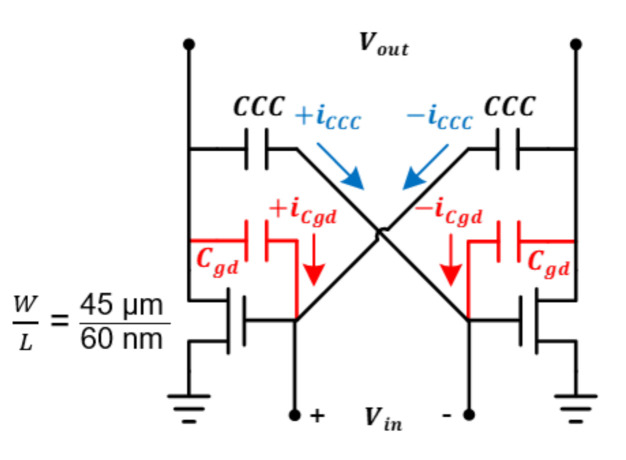
PA and DA circuits diagram.

**Figure 10 sensors-21-01563-f010:**
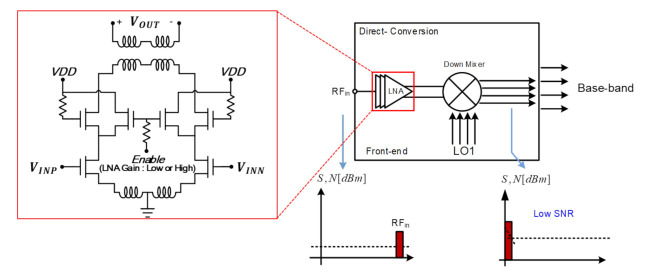
Direct conversion receiver architecture.

**Figure 11 sensors-21-01563-f011:**
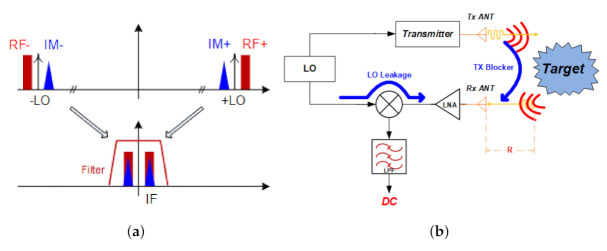
Disadvantage of direct conversion receiver architecture. (**a**) Image signal. (**b**) LO and TX leakage.

**Figure 12 sensors-21-01563-f012:**
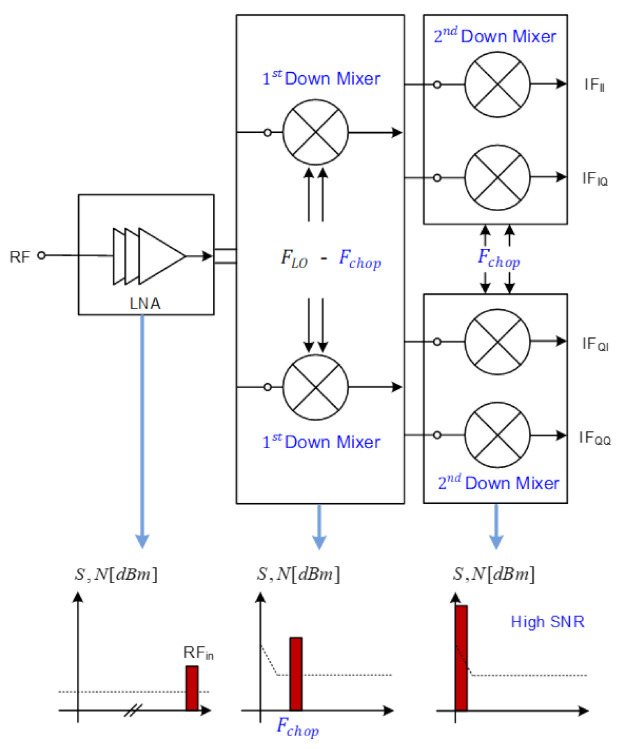
Double conversion receiver architecture.

**Figure 13 sensors-21-01563-f013:**
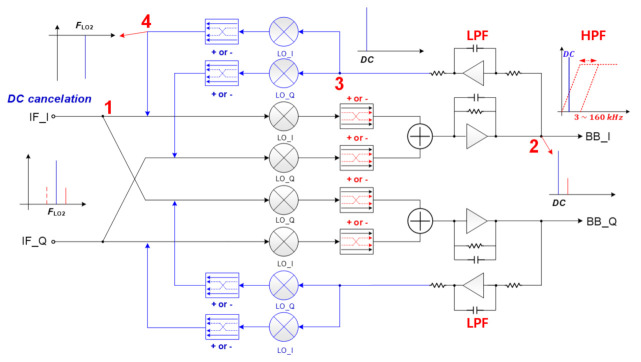
DC offset cancellation technique in the receiver.

**Figure 14 sensors-21-01563-f014:**
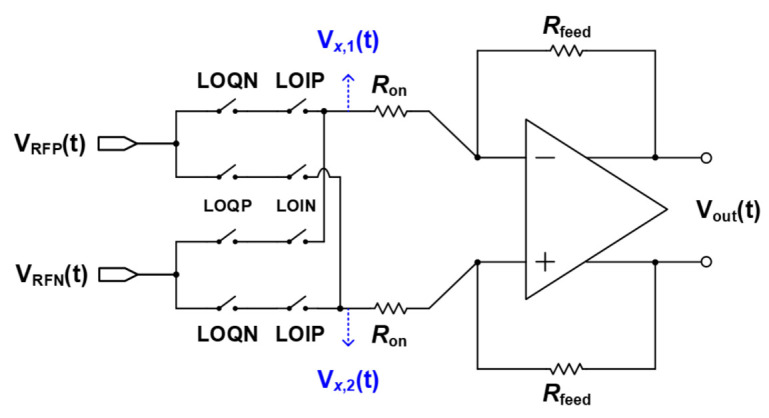
Mixer and Transimpedance Amplifier (TIA) circuit diagram in 25% duty cycle.

**Figure 15 sensors-21-01563-f015:**
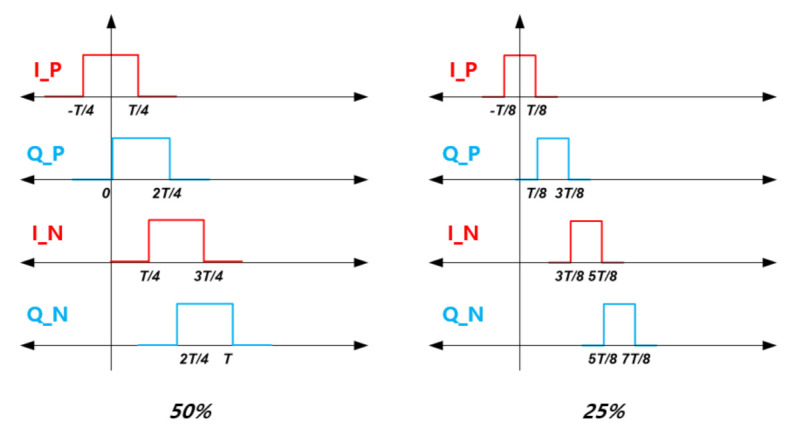
50% and 25% duty cycles for differential I and Q phases of the mixer.

**Figure 16 sensors-21-01563-f016:**
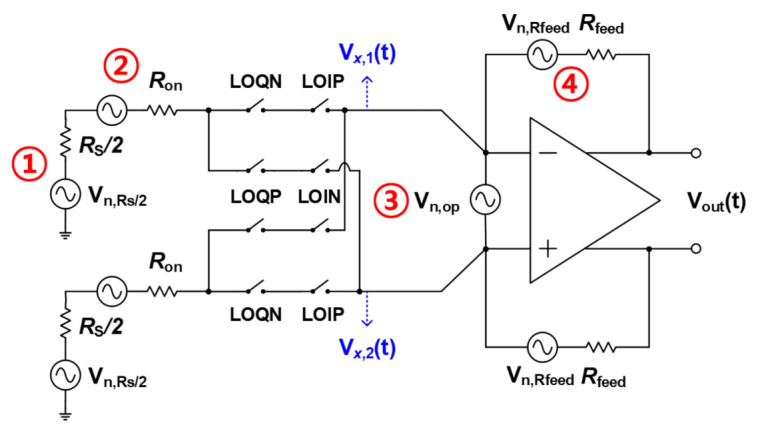
Equivalent noise analysis model of proposed mixer.

**Figure 17 sensors-21-01563-f017:**
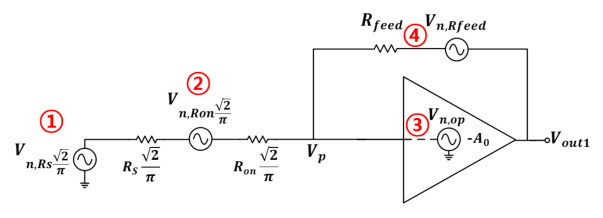
Equivalent noise analysis using half circuit modeling.

**Figure 18 sensors-21-01563-f018:**
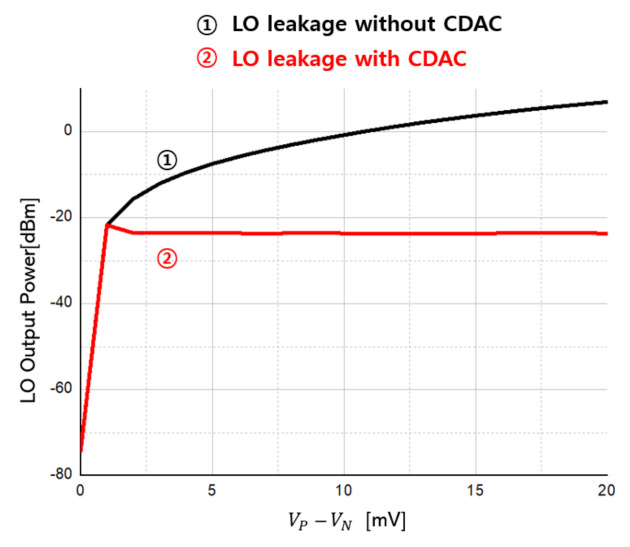
Simulation of LO leakage suppression by the proposed transmitter.

**Figure 19 sensors-21-01563-f019:**
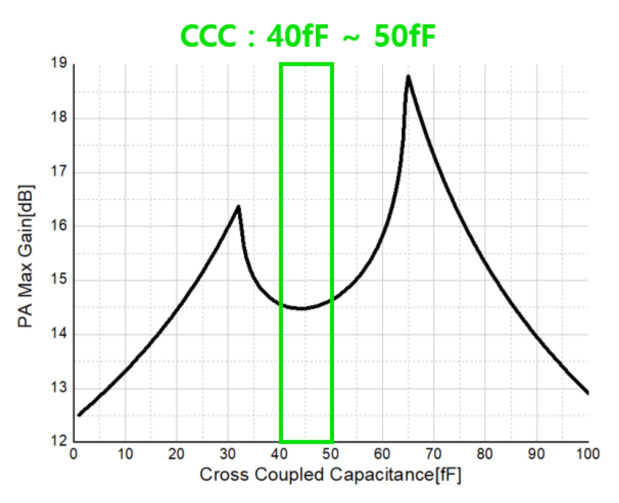
Maximum gain simulation according to CCC value as in Figure 9.

**Figure 20 sensors-21-01563-f020:**
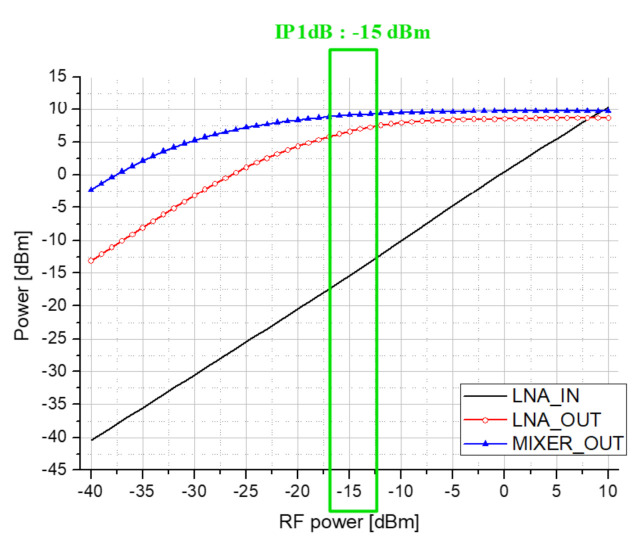
Front-end P1dB simulation result.

**Figure 21 sensors-21-01563-f021:**
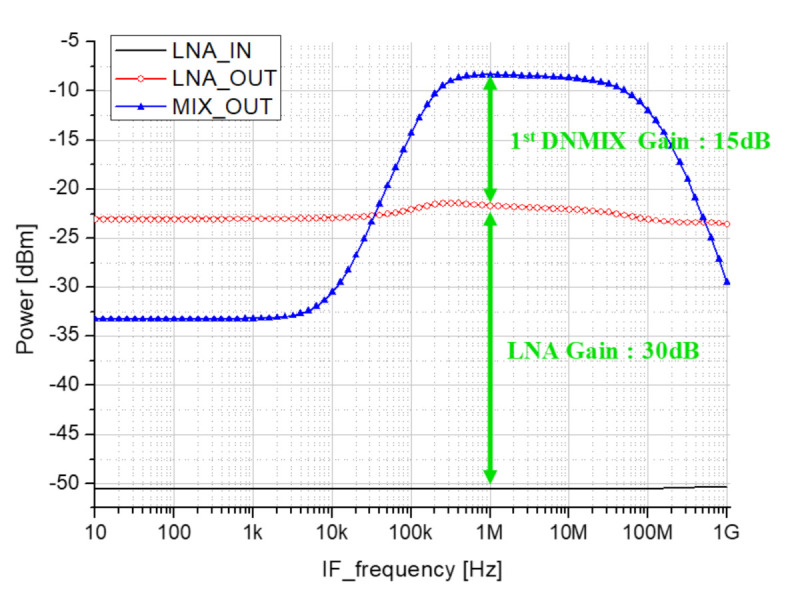
Front-end gain simulation result.

**Figure 22 sensors-21-01563-f022:**
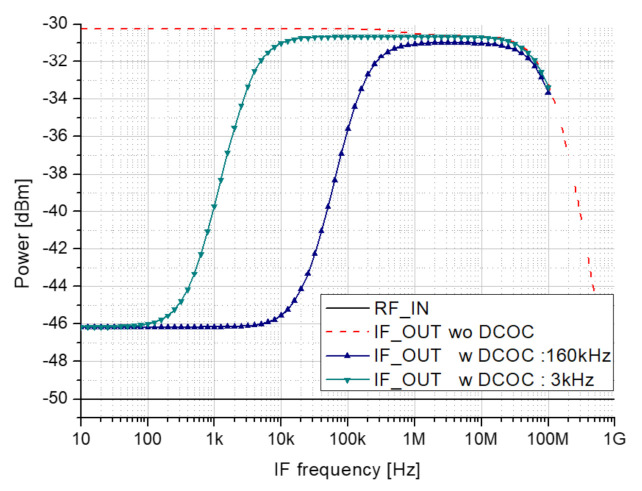
IF-stage gain simulation result.

**Figure 23 sensors-21-01563-f023:**
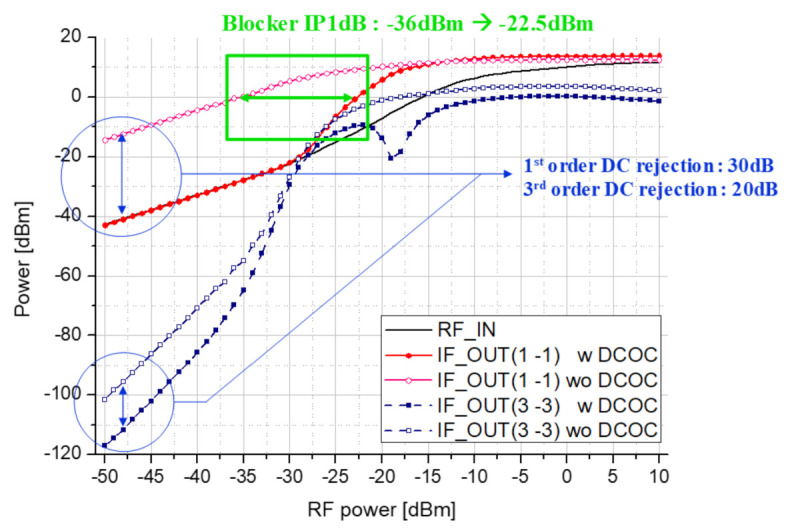
Blocker P1dB and DC rejection simulation result of proposed receiver.

**Figure 24 sensors-21-01563-f024:**
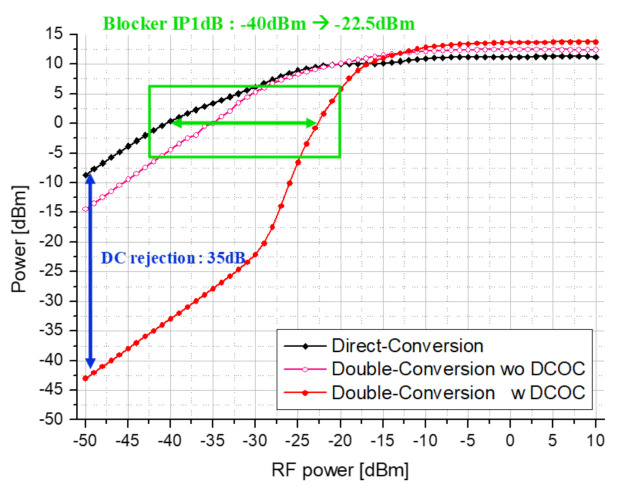
Simulation result of blocker P1dB and DC rejection for comparison.

**Figure 25 sensors-21-01563-f025:**
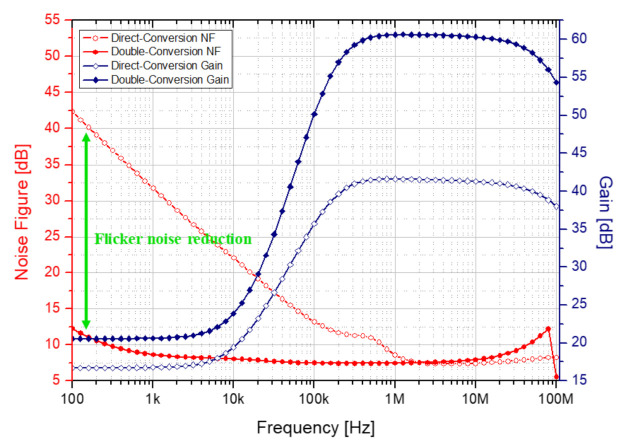
Noise simulation result for comparison.

**Figure 26 sensors-21-01563-f026:**
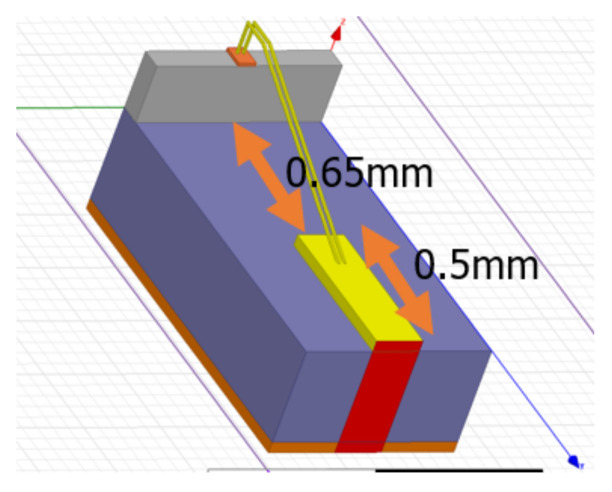
3D modeled double bond wire.

**Figure 27 sensors-21-01563-f027:**
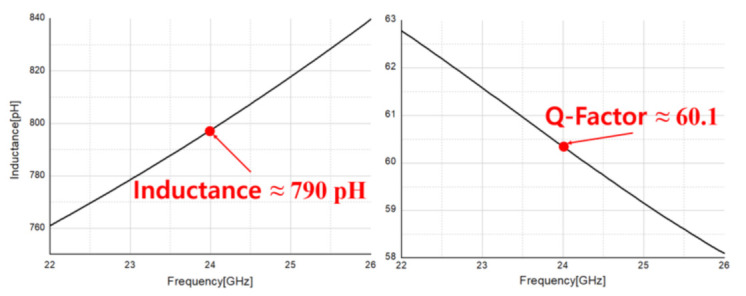
Simulated Q-factor and inductance of 3D modeled double bond wire.

**Figure 28 sensors-21-01563-f028:**
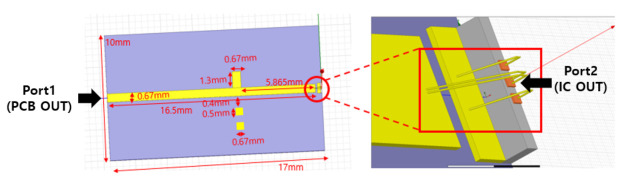
Wire bonding matching circuit applying stub circuit.

**Figure 29 sensors-21-01563-f029:**
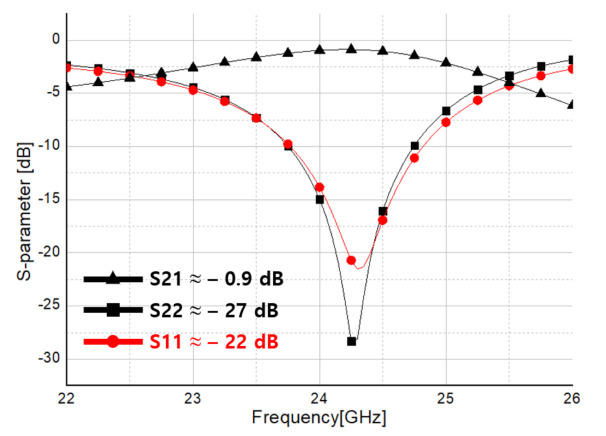
RF Front-End S-parameter simulation result including stub circuit.

**Figure 30 sensors-21-01563-f030:**
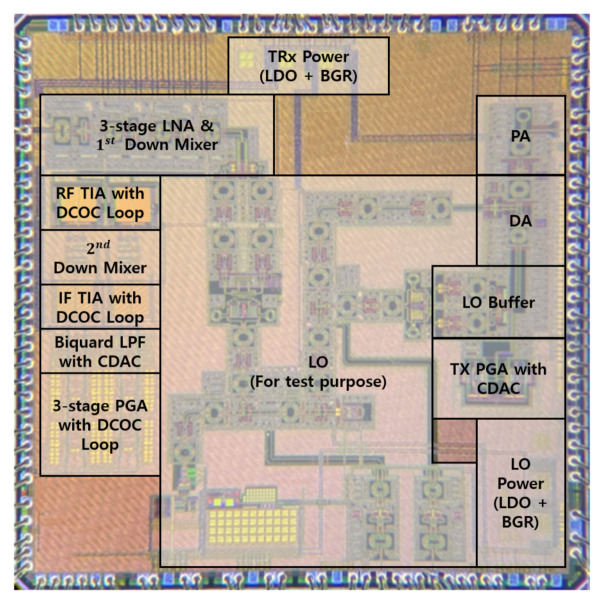
Transceiver die micrograph for RADAR System.

**Figure 31 sensors-21-01563-f031:**
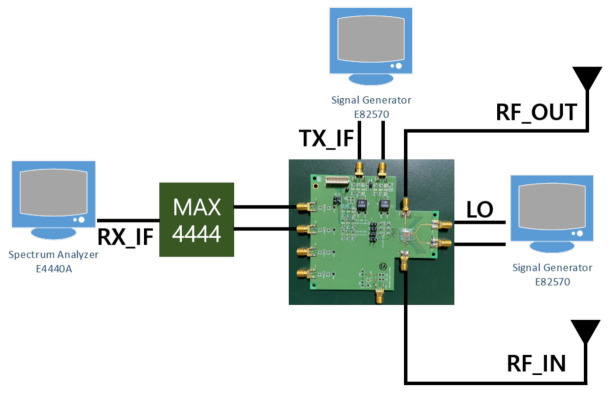
PCB and radar test environment.

**Figure 32 sensors-21-01563-f032:**
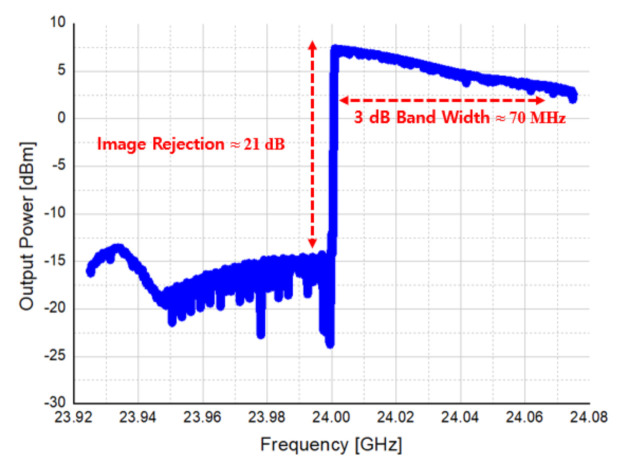
Measurement result of transmitter frequency response.

**Figure 33 sensors-21-01563-f033:**
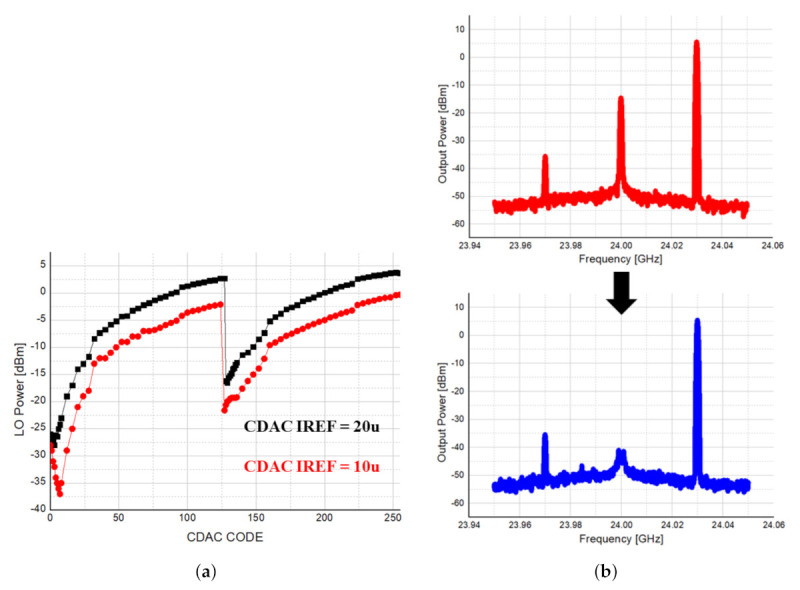
LO leakage power according to 8-bit CDAC code. (**a**) LO leakage according to 8-bit CDAC code; (**b**) the result of the output spectrum of the transmitter by applying the optimal code.

**Figure 34 sensors-21-01563-f034:**
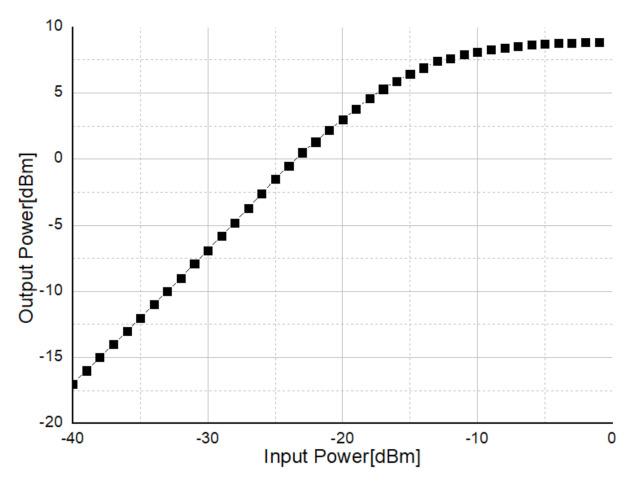
Measurement result of transmitter maximum output.

**Figure 35 sensors-21-01563-f035:**
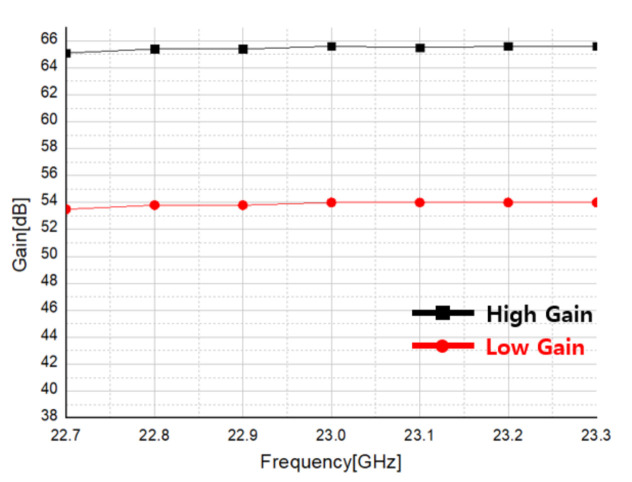
Bandwidth measurement result of the proposed receiver.

**Figure 36 sensors-21-01563-f036:**
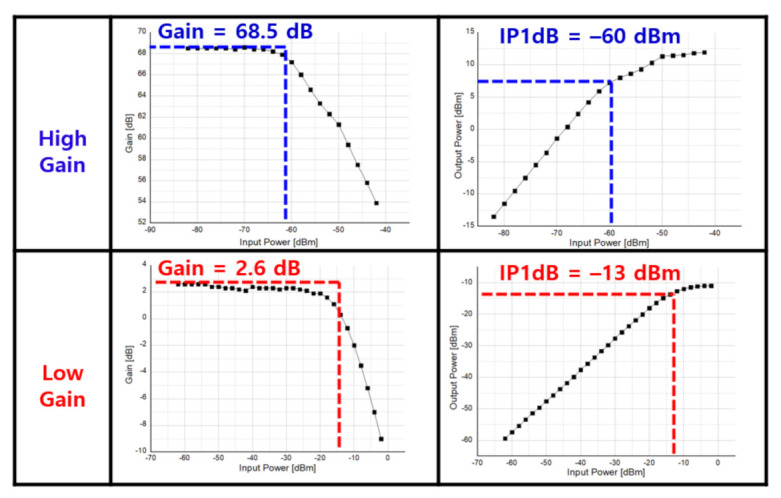
Receiver RF Front-End P1dB measurement result.

**Figure 37 sensors-21-01563-f037:**
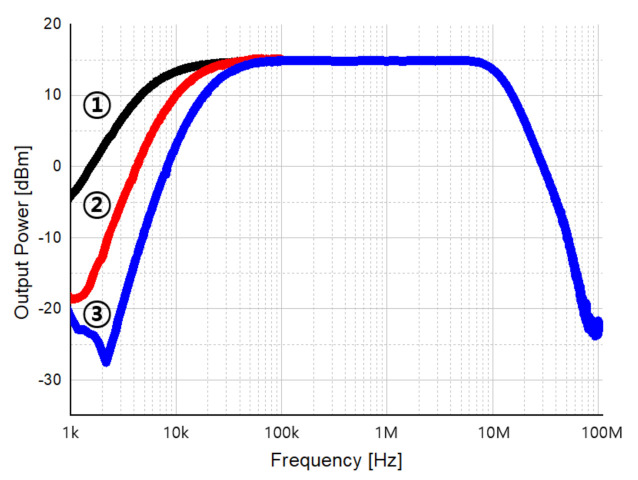
Measurement result of high-pass filter frequency response. Graph ① is the measurement result with only the DCOC of RF TIA on, and shows the cut-off characteristic of 20 dB/dec and 10 kHz. In the graph ②, two DCOCs were turned on, and the cut-off characteristics of 30 dB/dec and 20 kHz were confirmed. Finally, graph ③ was measured with three DCOCs turned on, and showed cut-off characteristics of 40 dB/dec, 20 kHz, and it was confirmed that the Doppler frequency can be detected properly.

**Figure 38 sensors-21-01563-f038:**
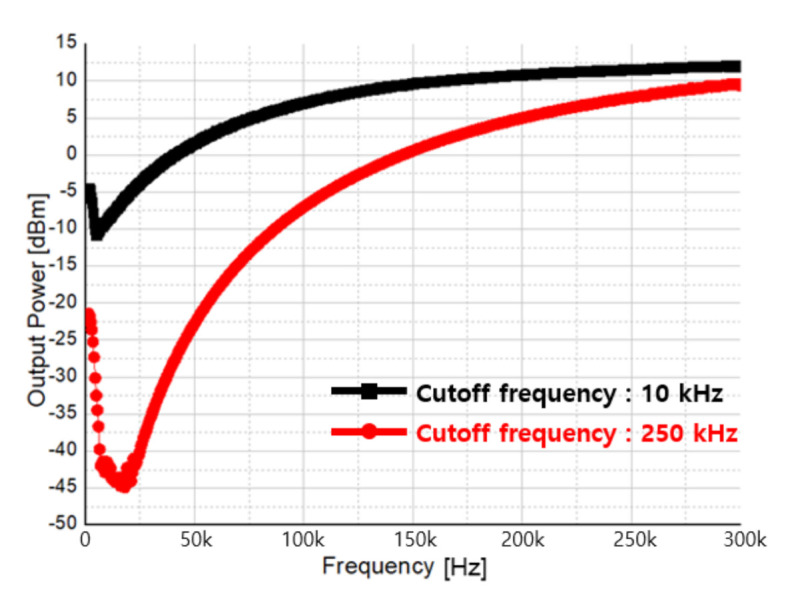
Measurement result of DCOC cut-off control.

**Figure 39 sensors-21-01563-f039:**
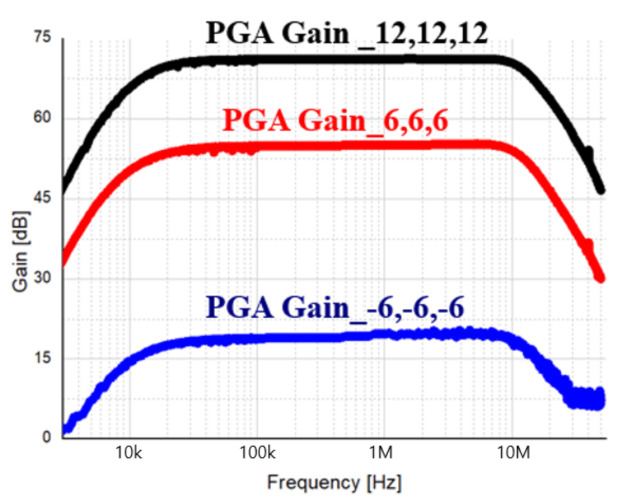
Measurement result of PGA gain of receiver.

**Figure 40 sensors-21-01563-f040:**
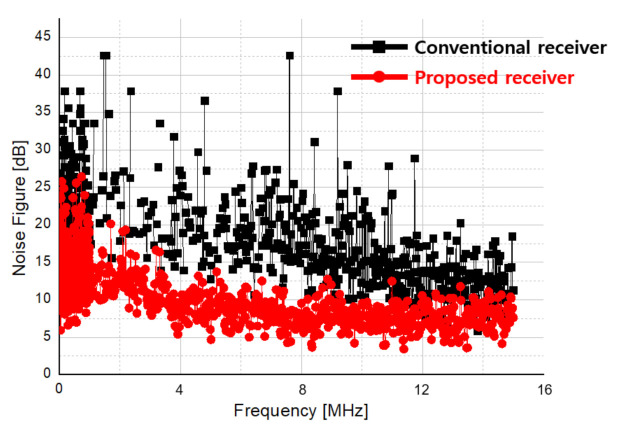
Noise figure result of conventional and proposed receivers.

**Figure 41 sensors-21-01563-f041:**
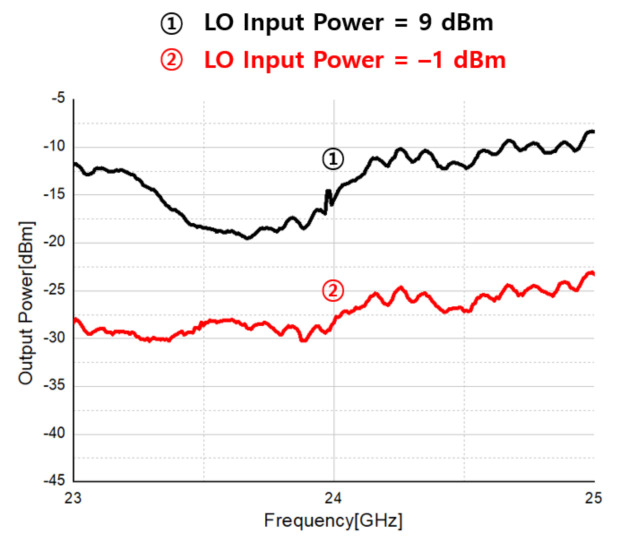
Measurement result of receiver LO leakage.

**Figure 42 sensors-21-01563-f042:**
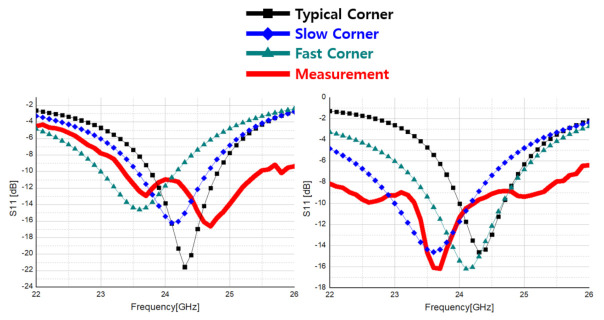
$S_{11}$ measurement and comparison with simulation: transmitter (left), receiver (right).

**Figure 43 sensors-21-01563-f043:**
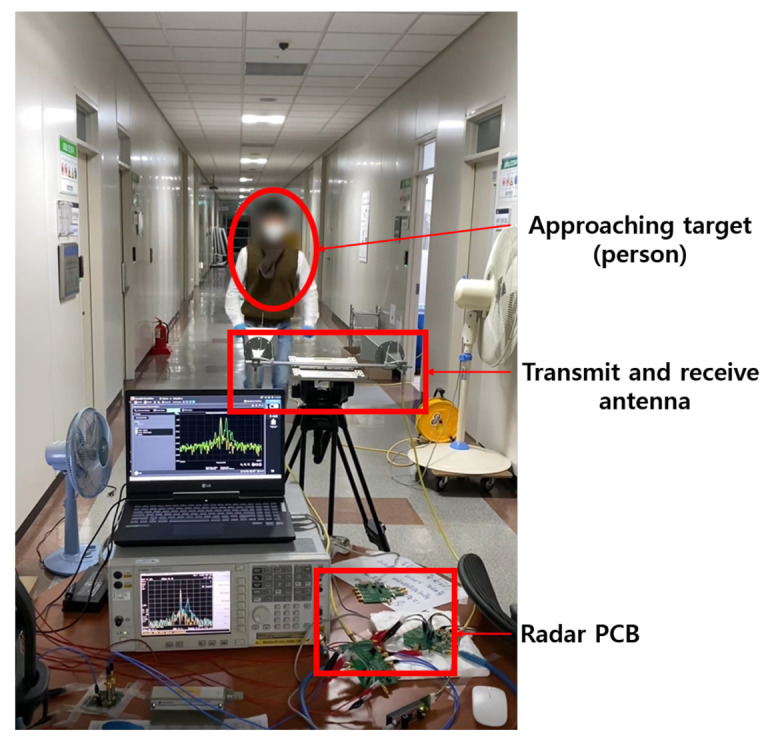
Measurement environment for radar test.

**Figure 44 sensors-21-01563-f044:**
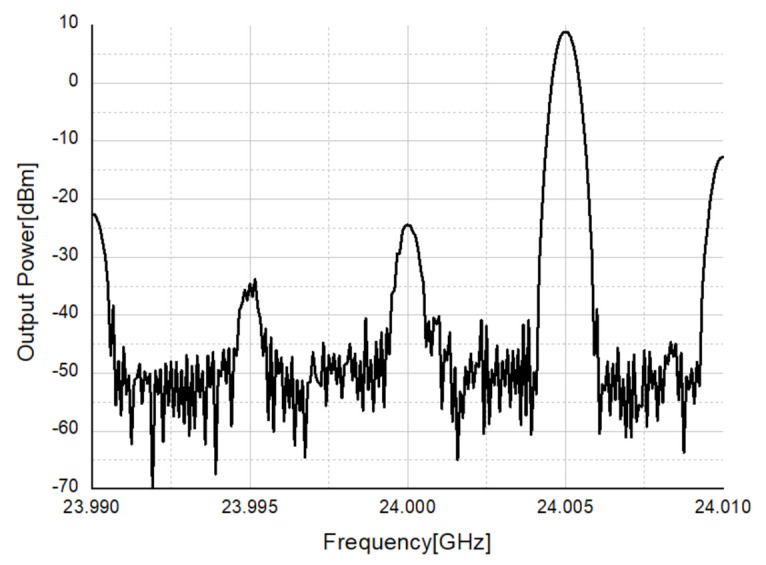
Transmitter output power measurement in CW radar.

**Figure 45 sensors-21-01563-f045:**
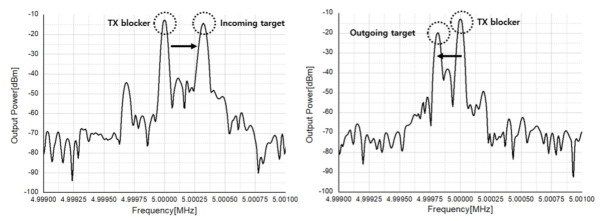
Measurement result of incoming (left) and outgoing targets (right).

**Figure 46 sensors-21-01563-f046:**
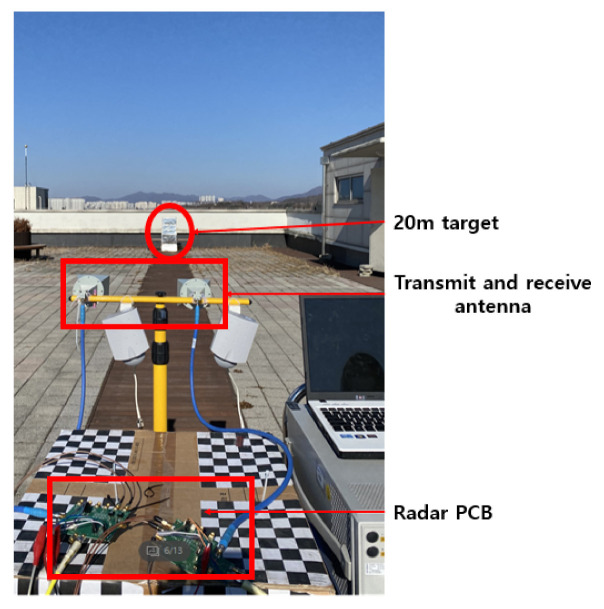
FMCW mode radar test environment.

**Figure 47 sensors-21-01563-f047:**
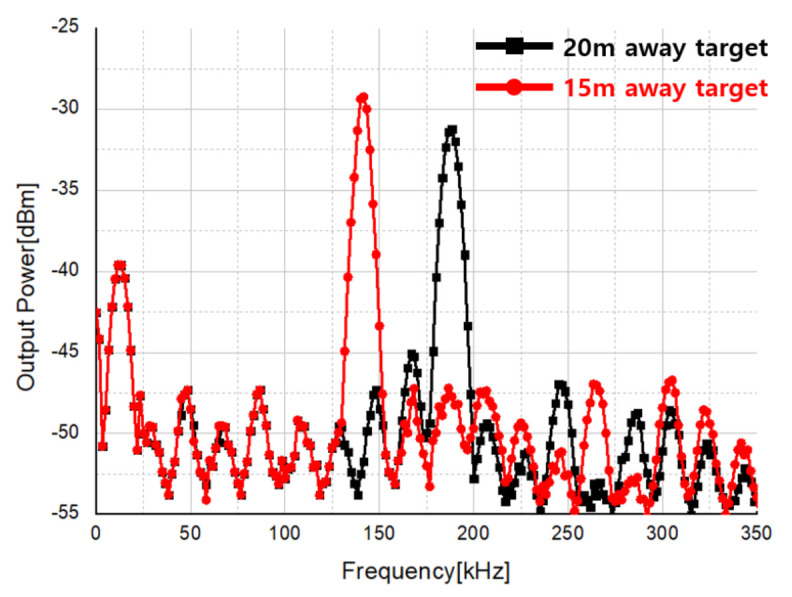
Measurement result of a stationary target using FMCW mode.

**Figure 48 sensors-21-01563-f048:**
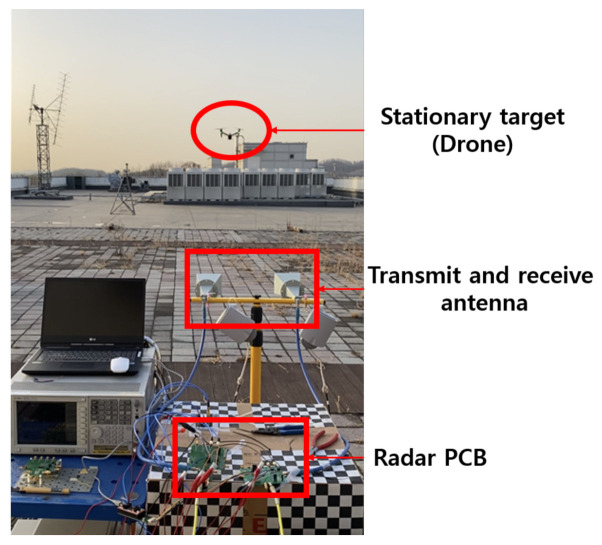
Pulse mode radar test environment.

**Figure 49 sensors-21-01563-f049:**
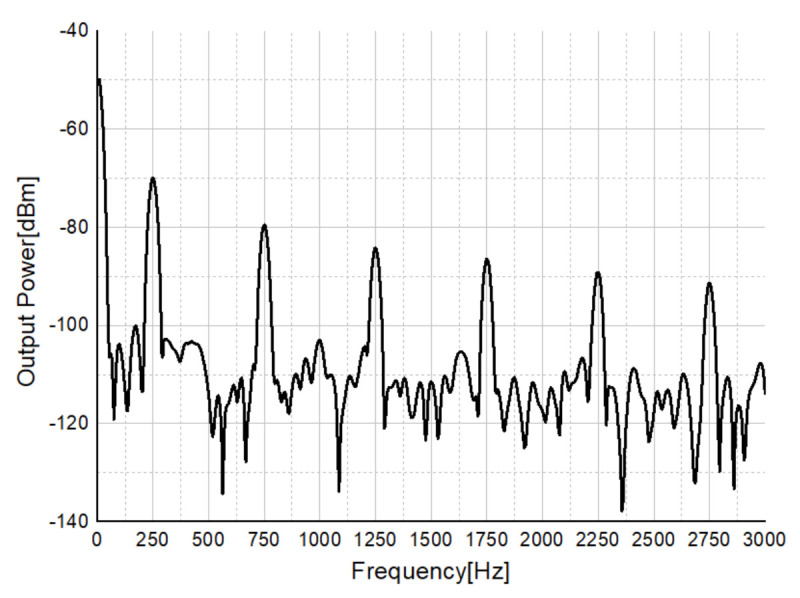
Measurement result of a stationary target using pulse mode with a hovering drone.

**Table 1 sensors-21-01563-t001:** Beat frequency calculation of FMCW radar (only UP chirp).

Velocity [km/h] Distance [m]	10	20	30	40	50	$f_{\max}$ (kHz)	$f_{\min}$ (kHz)
10	32.11	64.66	97.21	129.76	162.32	162.32	32.11
20	31.66	64.22	96.77	129.32	161.87	161.87	31.66
30	31.22	63.77	96.32	128.88	161.43	161.43	31.22
40	30.77	63.33	95.88	128.43	160.98	160.98	30.77
50	30.33	62.88	95.43	127.99	160.54	160.54	30.33
60	29.89	62.44	94.99	127.54	160.09	160.09	29.89
70	29.44	61.99	94.55	127.10	159.65	159.65	29.44
80	29.00	61.55	94.10	126.65	159.20	159.20	29.00
90	28.55	61.10	93.66	126.21	158.76	158.76	28.55
100	28.11	60.66	93.21	125.76	158.32	158.32	28.11

**Table 2 sensors-21-01563-t002:** Radar transceiver design requirements.

No.	Specification	Variable	Value	Unit
1	Maximum transmission power	$P_{T}$	13	dBm
2	RCS	σ	0.01	m${}^{2}$
3	Transmit antenna gain	$G_{T}$	25	dB
4	Receive antenna gain	$G_{R}$	25	dB
5	Center frequency	$f_{c}$	24	GHz
6	Transmit wavelength	λ	0.0125	m
7	Maximum thermal noise	$T_{e}$	−144	dBm/Hz
8	Frequency modulation bandwidth	B	75	MHz
9	Frequency modulation time	T	512	s
10	Transmitter bandwidth	-	75	MHz
11	Receiver bandwidth	-	75	MHz
12	Maximum noise figure	$NF_{max}$	12.9	dB
13	Minimum detectable input signal	$P_{in,min}$	−115.1	dBm
14	Maximum detection distance (pulse)	R	322	m
15	Maximum detection distance (FMCW)	R	84	m

**Table 3 sensors-21-01563-t003:** Transmitter design requirements.

Design Parameters	Unit	Value
Center frequency	[GHz]	24
PA&DA maximum gain	[dB]	14
PGA gain	[dB]	−6 ∼24 ${}^{a}$
OP1dB	[dBm]	7
Maximum output power	[dBm]	13
$S_{11}$ at TX output	[dB]	<−10
LO leakage rejection	[dB]	30
Bandwidth	[MHz]	75
Supply voltage	[V]	1.2
Current consumption	[mA]	<86

^*a*^ Controllable in 6 dB steps.

**Table 4 sensors-21-01563-t004:** Receiver design requirements.

Design Parameters	Unit	Value
Frequency	[GHz]	23.4∼26
Noise figure	[dB]	7.5∼12.5
3-stage LNA gain	[dB]	10∼30 ${}^{a}$
1st&2nd mixer gain (TIA included)	[dB]	14 ∼20
Baseband gain	[dB]	−18∼36 ${}^{b}$
Blocker IP1dB	[dBm]	−36∼−22.5
IIP3	[dBm]	−26∼−12.5
$S_{11}$(RF)	[dB]	<−30
HPF cutoff frequency	[kHz]	3 or 160
LPF cutoff frequency	[MHz]	10∼80
Chopping frequency	[MHz]	1.25∼40
DC rejection	[dB]	15∼30
Supply voltage	[V]	1.2
Current consumption	[mA]	120

^*a*^ Controllable in 10 dB steps, ^*b*^ Controllable in 6 dB steps.

**Table 5 sensors-21-01563-t005:** Summary and comparison of radar transceiver measurement results.

Spec.	Unit	This Work			[21]	[22]	[23]
		TX	RX				
Process	nm	65 CMOS	65 CMOS		180 CMOS	180 CMOS	180 CMOS
Architecture	-	Direct	Direct	Double	Double	Direct	Direct ${}^{a}$
Frequency	GHz	24	23	23	2.14	10.5	2.5
NF @100 Hz	dB	-	35	15	-	-	-
NF @100 kHz	dB	-	22	12	16	-	-
NF @1 MHz	dB	-	17	5	6.2	-	-
NFmin	dB	-	17	5	6.2	11.5	2.8
Gain	dB	23	66	68.5	44	−4.5 ${}^{b}$	62
IP1dB	dBm	−7	−54	−60	-	-	-
OP1dB	dBm	6.5	10	7	-	-	-
DC Rejection	dB	-	-	35	-	-	-
LO Leakage Rejection	dB	37	-	-	-	-	-
LO Leakage	dBm	−2.5	−28 ${}^{c}$		-	-	-
BW	GHz	0.07	2		0.003	-	0.02
Supply Voltage	V	1.2	1.2		1.8	1.5	1.2
Current	mA	86	99		15	24	14

^*a*^ only narrow band, ^*b*^ only front-end gain, ^*c*^ LO input power of −1 dBm.

**Table 6 sensors-21-01563-t006:** Summary of the radar test.

Mode	Variable	Value	Unit
pulse	Tx power	−20	dBm
	τ	2	ms
	PRF	250	Hz
	Detection range	20∼100	m
	Target	Drone, people	-
FMCW	Tx power	6.5	dBm
	B	75	MHz
	T	50	s
	R	25	m
	$f_{d}$	150∼400	kHz
	Detection range	15∼40	m
	Target	Metal square object	-
CW	Tx power	8.8	dBm
	Doppler Frequency	±300	Hz
	Target	Drone, people	-

## Data Availability

Data available on reasonable request from the corresponding author.

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
