# Peer review of "RF Transceiver for the Multi-Mode Radar Applications"

_sensors, 2021, doi:10.3390/s21051563_

Round 1
Reviewer 1 Report
The paper seems interesting but is quite difficult to read.
The mixture of figures doesn’t clarify the subject. The authors should supply the text block diagrams end then the electronic scheme.
The contribution of the authors is not clear. For example, what is their work on the LNA? Do they designed a new one or only associated 3 standard LNA.
Table 1 and Table 2 are useless and should be replace by a unique figure.
What is the benefit of figure 2?
The values in table 5 are not justified. For example, why an antenna gain of 25dB is required? The drone should be described to justify the -30 dB of RCS at 24 GHz.
Why the 13 dBm in table 5 becomes 12 dBm in table 6?
Figure 10 has a bad resolution.
The mixer described on figure 10 contains 8 interrupters, why? It seems only 4 will do the same function.
Equations from (9) to (19) should be significantly reduced.
What do the authors designed in the presented transceiver shown in figure 30?
Results presented in Figure 47: the FMCW mode seems not supply the desired response. Consider making a new measurements.
Consider reshape the paper in order to better described your work by focusing on the key points.
Reviewer 2 Report
This paper describes an innovative CMOS integrated circuit radar transceiver supporting pulse, FMCW and CW multi-mode operation. Efforts have been focused on reducing circuit flicker noise, LO leakage, DC offset to improve the receiver noise figure and target detection performance.
Comments,
- Some of the descriptions on pulse and FMCW radar parameters in Section 2 are not correct. Such as,
Equation 6, is for FMCW target range calculation, not range resolution. Therefore, values in Table 3 are also not range resolution. FMCW radar range resolution should be
Also for a FMCW radar, Doppler shift frequency only depends on the target velocity and it is necessary to have both up and down beat frequency to estimate the Doppler frequency shift. Values in Table 4 are only beat frequencies for up chirps, NOT Doppler frequencies.
- Please add discussion on the flicker noise sources, primarily introduced by the CMOS circuit or the LO?
- The paper uses “LO radiation” and “damage of the signal” in many places. Suggest to use “LO leakage” (since the unwanted LO leakage is due to insufficient RF-LO isolation within the mixer circuit, not through air radiation) and change “damage” to “contamination”.
- Line 69, “kTe is the thermal noise …” should be “KTe is the thermal noise power density …” (KTeBW is thermal noise power).
- Page 3, Eq(5), please give the bandwidth value for Pin,min and maximum detection range calculations.
- Table 5. Line 7, Te = -144K. Please verify this value.
- Figure 2, in the Tx chain, the Tx signal output is at 24.005 GHz, the received signal should be 24.005+fd GHz.
- Line 141-143, what is the RF-LO isolation for the Tx upconverter mixer? How come the LO leakage is larger than TX RF signal?
- Line 258-259, the paper claims that the noise figure of the mixer with 25 % dutycycle is significantly improved compared to that with a 50% dutycycle. Does that mean the radar Tx/Rx dutycycle should be kept under 25%? Will this limit the radar operation, such as pulse repetition frequency (PRF)?
- Table 7, “1st&2nd mixer gain” 14~20 dB. Conventional mixer is a passive component with conversion loss. Why the mixer here has positive gain values?
- Line 323, “… DCOC loop are added to lower noise figure of about 30 dB …”, should it be 20 dB?
- Figure 36, The values of the dashed lines for IP1dB, OP1dB are confusing. For instance, the top row figures (high gain), The left figure shows IP1dB = -60 dBm (but the vertical dashed line is at about -62 dBm), the right figure vertical dashed line is at ~ -56 dBm for OP1dB = 7 dBm. Similar for the bottom row figures (low gain).
- Figure 38, please add labels to the red and black curves.
- Figure 41, at which LO frequency the LO leakage was measured as -20 dBm and -33 dBm when LO input power were 9 dBm and -1 dBm, respectively? The red curve does not show -33 dBm point.
- Figure 42, the S11 parameters for transmitter and receiver are not at the same frequency, why?
Round 2
Reviewer 1 Report
The paper has been improved. I still find that the section 3 could be clarified. Nevertheless, it is acceptable for publication.